# Testing the assumptions in emergent constraints: Why does the 'Emergent constraint on equilibrium climate sensitivity from global temperature variability' work for CMIP5 and not CMIP6?

Mark S. Williamson[1,2], Peter M. Cox[1,2], Chris Huntingford[3], and Femke J. M. M. Nijsse[1,4]

[1]Global Systems Institute, Faculty of Environment, Science and Economy, University of Exeter, UK.
[2]Climate Dynamics, Department of Mathematics and Statistics, Faculty of Environment, Science and Economy, University of Exeter, UK.
[3]UK Centre for Ecology and Hydrology, Wallingford, UK.
[4]Department of Geography, Faculty of Environment, Science and Economy, University of Exeter, UK.

**Correspondence:** Mark S. Williamson (m.s.williamson@exeter.ac.uk)

**Abstract.** It was shown that a theoretically derived relation between annual global mean temperature variability and climate sensitivity held in the CMIP5 climate model ensemble (Cox et al. (2018a), hereafter CHW18). This so called emergent relationship was then used with observations to constrain the value of equilibrium climate sensitivity (ECS) to about 3$^o$C. Since this study was published, CMIP6, a newer ensemble of climate models has become available. Schlund et al. (2020) showed that many of the emergent constraints found in CMIP5 were much weaker in the newer ensemble including that of CHW18. As the constraint in CHW18 was based on a relationship derived from reasonable physical principles it is of interest to find out *why* it is weaker in CMIP6. Here, we look in detail at the assumptions made in deriving the emergent relationship in CHW18 and test them for CMIP5 and CMIP6 models. We show one assumption, that of low correlation and variation between ECS and the internal variability parameter, a parameter that captures chaotic internal variability as well as sub-annual (fast) feedbacks, while true for CMIP5 is not true for CMIP6. When accounted for, an emergent relationship appears once again in both CMIP ensembles implying the theoretical basis is still applicable although the original assumption in CHW18 is not. Unfortunately however, we are unable to provide an emergent constraint in CMIP6 as observational estimates of the internal variability parameter are too uncertain.

## 1 Introduction

Since the first general circulation climate models were introduced in the 1960s (Manabe and Bryan, 1969; Manabe and Wetherald, 1975) an ever increasing amount of effort has been spent developing and improving these models to produce simulations that are increasingly more realistic and feature more of the processes and interactions present in the real world. The progress and understanding of the processes governing the Earth's climate as a result has been impressive. However, even after decades of research, the range of predictions of some key characteristics of the Earth's future climate coming from these models are actually increasing rather than narrowing with time, one particular characteristic being the amount of warming due to doubling of $CO_2$ at equilibrium, known as equilibrium climate sensitivity (ECS, Sherwood et al. (2020)). Even though the latest state-

of-the-art climate models in the Coupled Model Intercomparison Project 6 (CMIP6, Eyring et al. (2016)) have a larger range of ECS values ($[1.84K\ 5.68K]$) than previous CMIP model ensembles (Forster et al., 2019), the latest IPCC estimates have actually narrowed. For decades the IPCC 'likely' range for ECS was between 1.5 to 4.5 K. In the latest report (IPCC, 2021) this was reduced to 2.5K to 4K with a best estimate of 3K.

There have been numerous attempts (Knutti et al., 2017) to constrain ECS using the historical warming record and paleo-climate data as well as climate model experiments. Researchers have also used the emergent constraint technique (Hall et al., 2019; Brient, 2020; Williamson et al., 2021) to constrain ECS (see for example Covey et al. (2000); Knutti et al. (2006); Masson and Knutti (2011a); Hargreaves et al. (2012); Sherwood et al. (2014); Caldwell et al. (2018) and the many references listed in Williamson et al. (2021)). The basic idea of emergent constraints is to identify an observable of the climate $x$ that varies significantly across a climate model ensemble and that exhibits a statistically significant relationship $f(x)$ with another variable $y$ describing an aspect of the climate model's future state. The relationship $y = f(x) + \varepsilon$, is referred to as an 'emergent relationship' where $\varepsilon$ is a relatively small departure from $f$. Since $x$ is observable, it can be measured in the real world. $f$ may then place a useful constraint on $y$, provided that the measurement uncertainty in $x$ is small compared to the range of simulated values. This constraint is 'emergent' because the emergent relationship $f$ cannot be diagnosed from a single climate model. It becomes apparent only when the full ensemble is analyzed.

There are pitfalls with the emergent constraint approach that must be guarded against particularly when the emergent relationships are not founded on well understood physical processes. For example, data-mining outputs from climate models could lead to spurious correlations (Caldwell et al., 2014) and less than robust constraints on future changes (Bracegirdle and Stephenson, 2013). Care is also needed drawing statistical inferences from ensembles of small numbers of models. The problem is compounded if models within the ensemble share common components giving a smaller effective ensemble size (Pennell and Reichler, 2010; Masson and Knutti, 2011b; Herger et al., 2018). Observations used to guide model development also may lead to dependencies (Masson and Knutti, 2012) and common structural inaccuracies (Sanderson et al., 2021).

One way of guarding against spurious correlations between $x$ and $y$ is to use analytical solutions of simplified models of the full complexity climate models to predict the emergent relationship $f$. $f$ can then be tested against the results from the complex models. This approach was used in Cox et al. (2018a) (CHW18) where the analytical solution of the one-box or Hasselmann model (Hasselmann, 1976) provided an emergent relationship between the statistics of historical global annual mean temperature variability ($x$) and ECS ($y$, see later for further details). This emergent relationship was tested and found to hold in the CMIP5 (Taylor et al., 2011) models although it was not without some debate regarding the applicability of the theory (Po-Chedley et al. (2018); Brown et al. (2018); Rypdal et al. (2018); Cox et al. (2018b), see section 3 for a discussion of these points). However, since these works were published, the newer CMIP6 ensemble has become available. Schlund et al. (2020) showed that many of the emergent constraints found in CMIP5 were much weaker in the newer ensemble including that of CHW18.

As the constraint in CHW18 was based on a relationship derived from reasonable physical principles it is of interest to find out *why* it got weaker in CMIP6. Some possible reasons are:

- The simple theory is not applicable to climate models and the real world. However, simple models (particularly two-box models) are regularly used to reproduce the annual global mean temperature response of climate models and they do it well (see Caldeira and Myhrvold (2013); Geoffroy et al. (2013b, a); Gregory (2000); Held et al. (2010); MacMynowski et al. (2011)).

- Estimates of the temperature variability observable ($x$) are uncertain enough to mask the relationship with ECS ($y$). This is unlikely as historical observations are long ($> 100$ yrs) and relatively un-autocorrelated in time (a few years) giving good estimators of the true values.

- The assumptions made in deriving the emergent relationship that held for CMIP5, no longer hold for CMIP6. This is something we test in this manuscript.

The central interest of this manuscript is to test the assumptions that go into the derivation of the emergent relationship in CHW18. These assumptions are outlined in section 3 and then tested in the CMIP5 and CMIP6 model ensembles with the aim of understanding why the emergent relationship in CHW18 is weaker for the CMIP6 model ensemble. Of course all assumptions will be ultimately wrong if perfect agreement is expected (the often used quote 'all models are wrong' applies). However, 'some models are useful' and we look for agreement 'for all practical purposes (FAPP)', a term coined by John Bell (Bell, 1990). We will largely not be interested in the final step of obtaining the emergent constraint that results from combining the emergent relationship with observations for reasons we will outline later in the manuscript.

The structure is as follows: In section 2 we review the methodology of CHW18 and how it is used in this study. In section 3 we explicitly list, discuss and test the assumptions in CHW18 and show which assumption fails for the CMIP6 model ensemble. In section 4 we show how to recover a robust emergent relationship in both CMIP5 and CMIP6 ensembles by including the forcing parameter in the the predictor $x$. In section 6 we make a rigorous test of the emergent relationship theory by numerical simulation and show it does a reasonable job (FAPP) reproducing the results seen in each of the ensembles of the full complexity CMIP climate models. We discuss and conclude in section 7.

## 2 CHW18 methodology

The response of the global mean surface air temperature anomaly $T(t)$ with time $t$ to forcing $Q(t)$ is assumed to be well modelled by the one-box or Hasselmann model (Hasselmann (1976), hereafter H76) in CHW18. Forcing in this model comes from random, short timescale weather noise as well as other external sources such as solar radiation and changes in greenhouse gas concentrations. Air temperature sensitivity to forcing is parameterized by $\lambda$, a term that lumps all the effects of the Earth system's feedbacks together. The single box has heat capacity $C$. In this model, $T(t)$ evolves according to

$$C\frac{dT}{dt} = Q(t) - \lambda T(t). \tag{1}$$

Solving this model results in a linear relation between ECS and a metric of temperature variability $\Psi$, which is a form of a fluctuation-dissipation theorem (Kubo, 1966; Leith, 1975). Explicitly

$$ECS = \sqrt{2} \frac{Q_{2 \times CO2}}{\sigma_Q} \Psi. \tag{2}$$

Where $Q_{2 \times CO2}$ is the radiative forcing resulting from doubling the atmospheric $CO_2$ concentration and $\sigma_Q$ is the standard deviation of a zero mean white noise process designed to model the fast (sub-annual), chaotic weather forcing on the slower Earth system components. $\Psi$ can be measured from temperature observations and is defined as

$$\Psi = \frac{\sigma_T}{\sqrt{-\log \alpha_{1T}}}, \tag{3}$$

where $\sigma_T$ is the standard deviation and $\alpha_{1T}$ is the autocorrelation at 1 year lag of annual global mean temperature. Details of this derivation can be found in CHW18 and Williamson et al. (2018).

CHW18 calculated the pair of values $(\Psi_i, ECS_i)$ for each of the $n = 16$ CMIP5 climate models labelled by $i \in \{1, 2, ..., n\}$ performing a simulation of the historical period 1880-2016. Plotting the $n$ pairs confirmed the theoretically expected $\Psi$ vs $ECS$ linear 'emergent relationship' with good correlation ($r = 0.77$, $r$ in this manuscript denotes Pearson's correlation coefficient). Combining this resulting emergent relationship with $\Psi$ from observational records of the same period gave an emergent constraint on ECS of $2.8 \pm 0.6^o$C (plus minus values are 66% confidence intervals).

Although there were more CMIP5 models available than the $n = 16$ used in CHW18, the choice of one model per modelling centre was made to avoid biasing the emergent constraint towards similar models. Where multiple models were available from the same centre, the model with the lowest root mean square error to the observational temperature record was chosen. Po-Chedley et al. (2018) and Schlund et al. (2020) repeated the analysis of CHW18 including these additional models and thus had a larger CMIP5 ensemble (larger $n$). They found the emergent relationship got slightly weaker, although it was still 'highly significant' in the language of Schlund et al. (2020).

In this manuscript we use the CHW18 methodology (further detailed in the original manuscript) and apply it to CMIP5 and CMIP6 models with the following differences: Here we look at the historical period 1880-2005 for both CMIP5 and CMIP6 ensembles following Schlund et al. (2020) rather than 1880-2016 as in CHW18. This is because the standard CMIP5 historical experiment ends in 2005. (Increasing the time period to present day by concatenating with one of the CMIP rcp or ssp future projection experiments slightly increases the strength of the correlation in the emergent relationship.) We also use a different ensemble of 15 CMIP5 models corresponding with those analyzed in Geoffroy et al. (2013b). Geoffroy et al. (2013b) also lists FGOALS-s2 however we leave this model out as it does not have a historical simulation with which to calculate $\Psi$. We use the Geoffroy et al. (2013b) ensemble as their published parameter values are used in section 6 to run simulations of box models. These simulations are used to compare the theory with the full complexity CMIP5 models. To make a fair comparison limits us to the same set. For the CMIP6 ensemble we use all models that have the necessary simulations for our analysis (piControl, historical and abrupt-4xCO2), a set of $n = 33$ models. For both CMIP ensembles we use one run for each model, preferably the one labelled or r1i1p1 (CMIP5) or r1i1p1f1 (CMIP6) where it exists. We look at the results for different runs of the same model in section 5 although we find no qualitative changes to the findings with the r1i1p1 (CMIP5) or r1i1p1f1 choices. A list of models used and their parameter values is given in appendix B.

**Figure 1.** $\Psi$ vs ECS emergent relationships in the CMIP5 (left panel) and CMIP6 (right panel) model ensembles running the historical experiment. The period 1880-2005 of each model's timeseries is used to calculate $\Psi$. ECS is determined from the abrupt4xCO2 experiment using the standard Gregory plot method. Individual models are plotted as circles (CMIP5 models are blue and CMIP6 models are red). The best fit line in the ordinary least squares sense is shown in black along with the standard deviation of the prediction error (black dotted line). Pearson correlation $r$ and $p$ value are given for each emergent relationship in each subplot title.

The $(\Psi, ECS)$ emergent relationships for CMIP5 and CMIP6 ensembles are shown in figure 1. The CMIP5 ensemble shows good correlation between $\Psi$ and ECS, $r(\Psi, ECS) = 0.66$, however for CMIP6 this is weaker, $r(\Psi, ECS) = 0.31$, confirming the results of CHW18 (although with slightly different historical period and set of CMIP5 models) and Schlund et al. (2020) (CMIP6).

Schlund et al. (2020) use the following definitions for significance based on $p$ value: An emergent relationship is called 'highly significant' if $p < 0.02$, 'barely significant' if $0.02 \leq p < 0.05$, 'almost significant' if $0.05 \leq p < 0.1$ and 'far from significant' if $p \geq 0.1$. We adopt their definitions in this manuscript. We find the $(\Psi, ECS)$ emergent relationship highly significant for CMIP5 and almost significant for CMIP6.

## 3 Assumptions in CHW18

The following assumptions are made in the CHW18 methodology to obtain the emergent relationship between $\Psi$ and ECS:

**A1** The $T(t)$ response to $Q(t)$ is modelled well by H76 for timescales greater than one year and less than the detrending window length (55 years in CHW18).

**A2** H76 is solved with a random, white noise forcing $Q(t)$ of zero mean and standard deviation $\sigma_Q$. This is designed to parameterize internally generated variability (from weather for example, Hasselmann (1976)). It is assumed that the response from all other sources of forcing in the historical period such as (but not limited to) GHGs, solar irradiance and volcanoes can be removed via detrending to a good approximation so that equation 2 applies to this period in both observations and CMIP models.

**A3** The forcing parameters, $Q_{2 \times CO2}$ and $\sigma_Q$ in equation 2, are uncorrelated to ECS and their variation is small relative to the variation in $\Psi$. This requirement makes $\Psi$ a good predictor of ECS.

There are further assumptions concerning the quantification of sources of uncertainty (structural, observational etc) in deriving the emergent constraint in CHW18. These are considered in more detail in Williamson and Sansom (2019) and Williamson et al. (2021). However, as we look only at the emergent relationship here, these will not discussed further.

### 3.1 Testing the assumptions

To summarise this subsection, assumption **A3**, is violated for the CMIP6 models. However, all other assumptions still apply FAPP for CMIP5 and CMIP6. In particular, it is the assumption of no correlation between ECS and the forcing parameter, $\sigma_Q$, that is no longer true for CMIP6. In CMIP6 significant correlation exists. Each assumption in order is discussed below.

Assumption **A1** was studied in detail in Williamson et al. (2018) and Cox et al. (2018b). To summarize, H76 only really has any physical justification when the timescales of interest are dominated by the well-mixed atmosphere and ocean surface layer (a few years to decades). It is well known H76 does a poor job of reproducing $T(t)$ on longer timescales (see e.g. Caldeira and Myhrvold (2013); Schwartz (2007, 2008); Foster et al. (2008); Kirk-Davidoff (2009); Knutti et al. (2008); Scafetta (2008)). This led some to question the use of H76 in CHW18 e.g. Rypdal et al. (2018). However, one can show analytically (Williamson et al., 2018) that a near-linear emergent relationship is also expected between ECS and $\Psi$ for the more realistic and widely used two-

box (Gregory, 2000; Held et al., 2010) and diffusion models (MacMynowski et al., 2011). Both two-box and diffusion models are known to do a good job of reproducing the global annual mean temperature response of CMIP climate models (Caldeira and Myhrvold, 2013; Geoffroy et al., 2013b). As the $T(t)$ solutions of CMIP6 models qualitatively have the same structural form as CMIP5 models to stepped and linearly increasing forcing (abrupt-4xCO2 and 1pctCO2 experiments respectively) we expect that two-box and diffusion models also emulate the CMIP6 models well. We fit two-box models to the CMIP6 ensemble (as Geoffroy et al. (2013b) did for CMIP5) later in the manuscript and can confirm this is indeed the case. The reason the $\Psi$ vs ECS linear relationship still holds to a good degree in the more complete two-box and diffusion models is because $\Psi$ is a statistic that is dominated by fast timescale processes of a few years, a feature H76 does capture well.

**A2** assumes the response to all external forcing (GHGs, volcanoes, etc) in the historical period can be removed to a good approximation by linearly detrending $T(t)$ in a 55 year moving window, leaving just the internally generated random variability parameterized as the response to random 'forcing' in H76.

This was the procedure introduced in CHW18 and we continue with the same procedure here for consistency and comparison. The reasons for using a 55 year window have been discussed in the original paper (Cox et al., 2018a) as well as subsequent publications (Cox et al., 2018b; Williamson et al., 2018). The reason for linear detrending is to remove the response due to the slow timescale in the climate. It turns out that when fitting two-box models to the CMIP models, a fast timescale ($\sim 4$ years) and a slow timescale ($\sim 200$ years) response result, see Geoffroy et al. (2013b) for example or the tables in the appendix of this manuscript. Linear detrending with a 55 year timescale fits nicely between the short and fast timescale and removes the slow response component. It also minimizes the uncertainty in the resulting emergent constraint (Cox et al., 2018a). Removing the slow timescale response leaves a signal that is more H76 (one-box) model like and therefore more like the underlying simple theory of the emergent relationship.

Assumption **A2** is to make the derivation of equation 2 (which is a derivation that applies to the piControl experiment) applicable to the historical simulations. Several works (Po-Chedley et al., 2018; Brown et al., 2018) showed this assumption to be false. In particular they showed that the detrending procedure in CHW18 does not remove the response to all external forcing. They also showed that better methods of removing forced variability slightly weakened the emergent relationship. Cox et al. (2018b) acknowledged this to be true, however they also showed that external forcing, provided it is common for all models in the ensemble, would actually be helpful and improve the emergent relationship. This was demonstrated using an ensemble of H76 and two-box models tuned to mimic the CMIP5 models running a variety of experiments with and without common and random forcing. A sketch of the reason is as follows: $\Psi$ is linearly proportional to sensitivity and given an ensemble of models with a range of sensitivities, more sensitive models will respond with a larger $\Psi$ (or response) if all models in the ensemble are given the same (common) forcing, providing a natural way of ordering the model's sensitivities. The common forcing in the historical simulations comes from volcanoes, anthropogenic trends, solar cycles etc.

Equation 2 predicts a linear relationship between ECS and $\Psi$ provided $Q_{2\times CO2}$ and $\sigma_Q$ can be treated as 'constants' across the model ensemble (assumption **A3**). A looser definition of 'constant' for equation 2 is stated in **A3**. In figure 2 (a) we plot $Q_{2\times CO2}$ against ECS and compute their correlation in both CMIP5 and CMIP6 ensembles. For both ensembles $Q_{2\times CO2}$ is uncorrelated to ECS ($r = -0.17$ for CMIP5 and $r = -0.07$ for CMIP6, both $p$ values, $p \geq 0.1$, are far from significant).

$Q_{2 \times CO2}$ is determined in the standard way for each model running an abrupt-4xCO2 experiment via a Gregory plot (Gregory et al., 2004).

In figure 2 (b) we plot the other forcing 'constant' $\sigma_Q$ against ECS. $\sigma_Q$ is estimated from the detrended temperature residual of each climate model's historical run. The standard deviation of white noise forcing $\sigma_Q$ is fitted for each model from the global annual mean temperature timeseries. This timeseries is linearly detrended with a rolling 55 year window. This is to isolate the $T(t)$ response to internal variability, analogous to how $\Psi$ is determined in the CHW18 methodology, to leave the noisy $T(t)$ response to white noise with standard deviation $\sigma_T$. The theoretical formula is given by (see Williamson et al. (2018) for example)

$$\sigma_T^2 = \frac{\sigma_Q^2}{2\lambda C}. \tag{4}$$

We rearrange this relation to get $\sigma_Q$ in terms of the observable $\sigma_T$ and the parameters $\lambda$ and $C$ (given in tables B1, B2, B3 and B4, see section 6 for details on how the H76 model parameters are fitted). Values of $\sigma_Q$ in both historical and piControl runs are also reported in these tables.

Consistent with CHW18, $\sigma_Q$ is uncorrelated to ECS in CMIP5 ($r = -0.09$), however in CMIP6 there is highly significant anti-correlation ($r = -0.58$, $p < 0.001$). We could equally estimate $\sigma_Q$ from piControl simulations. We choose the historical experiment for consistency with estimation of $\Psi$. Whichever simulation is used, the correlation with ECS remains largely invariant (piControl $r(\sigma_Q, ECS) = -0.09$ and $r(\sigma_Q, ECS) = -0.58$ in CMIP5 and CMIP6 respectively).

When plotting the combination of 'constants', $Q_{2 \times CO2}/\sigma_Q$, multiplying $\Psi$ in equation 2, (figure not shown), CMIP6 still has highly significant correlation between $Q_{2 \times CO2}/\sigma_Q$ and ECS ($r = 0.74$, $p < 0.001$). CMIP5 shows some, although far from significant anti-correlation ($r = -0.21$, $p = 0.46$).

## 4 Recovering an emergent relationship

We have confirmed that $\Psi$ is a good predictor of ECS for CMIP5 models although not for CMIP6 models. In figure 3, when $\sigma_Q$ is included in the $x$ axis predictor variable, a good emergent relationship is recovered for both CMIP ensembles, both have highly significant $p$ values of $p < 0.001$. One can also include $Q_{2 \times CO2}$ (although it is uncorrelated to ECS in both CMIP ensembles) in the predictor i.e. $ECS \propto \frac{Q_{2 \times CO2}}{\sigma_Q}\Psi$ to get a similarly skillful emergent relationship (figure not shown). We restrict to $\Psi/\sigma_Q$ as minimal degrees of freedom are preferred.

Where does the skill in predicting ECS using $\Psi/\sigma_Q$ come from? In CMIP5, it came from $\Psi$ (an observable). There is no skill in $\sigma_Q$ (it is uncorrelated with ECS). In CMIP6 the converse is roughly correct: There is limited correlation with ECS from $\Psi$ but good correlation from $\sigma_Q$, which, to our knowledge, is unfortunately not directly observable.

Theoretically, these findings should hold equally well in the piControl run, although the emergent relationships should have slightly weaker correlation for reasons outlined in section 3.1 and Cox et al. (2018b). Again, we find this is roughly true (see figure 4). For the piControl experiments we analyze the longest common period simulated in the CMIP5 and CMIP6 ensembles which is 200 years.

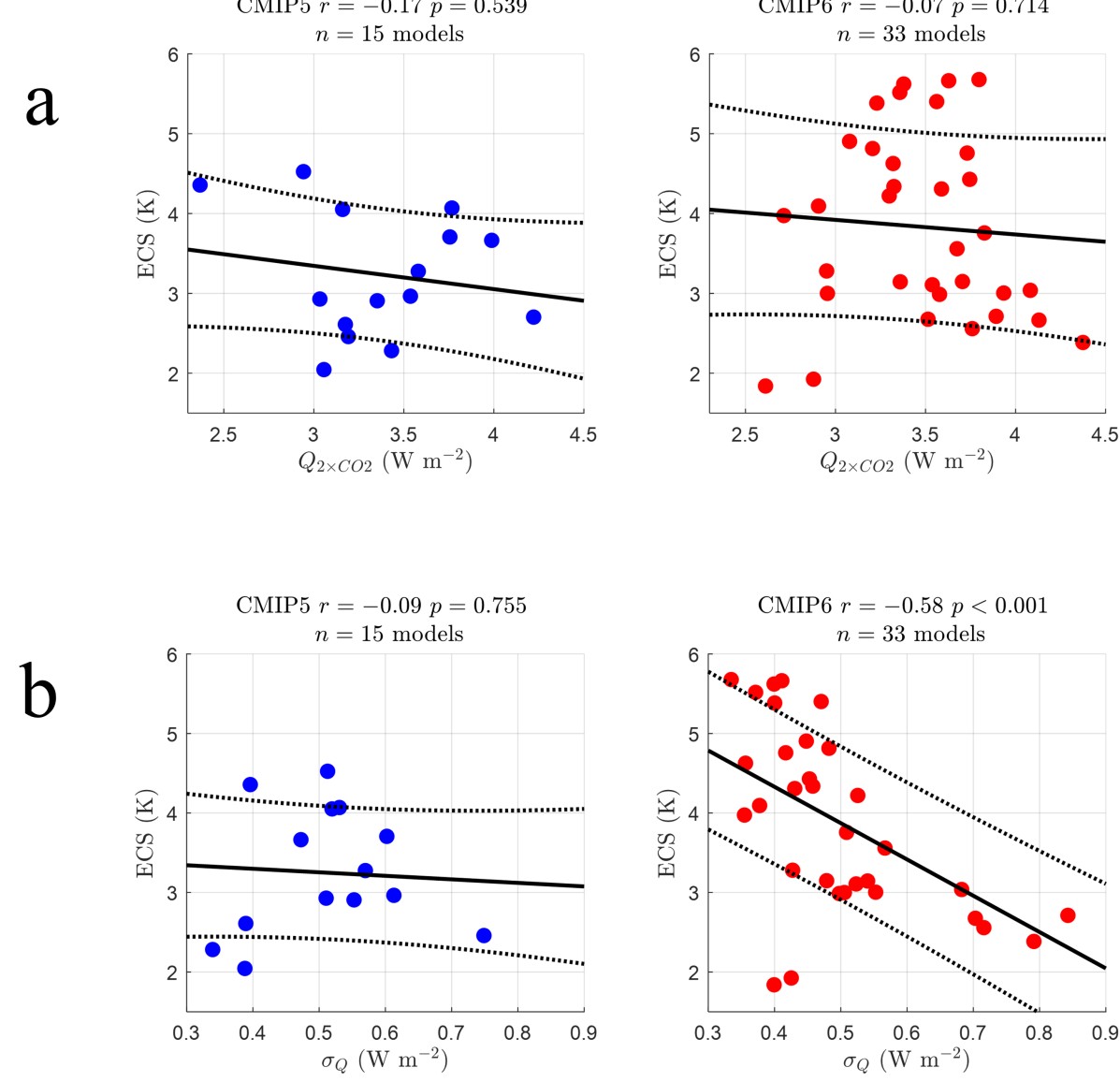

**Figure 2.** Individual models are plotted as circles (CMIP5 models are blue and CMIP6 models are red). The best fit line in the ordinary least squares sense is shown in black along with the standard deviation of the prediction error (black dotted line). Pearson correlation $r$ and $p$ value are given for each emergent relationship in each subplot title. (a) $Q_{2\times CO2}$ against ECS in the CMIP5 (left panel) and CMIP6 (right panel) model ensembles. $Q_{2\times CO2}$ is inferred from the abrupt4xCO2 experiment using the standard Gregory plot method. (b) $\sigma_Q$ against ECS in the CMIP5 (left panel) and CMIP6 (right panel) model ensembles. $\sigma_Q$ is calculated from the period 1880-2005 of each model's historical experiment timeseries.

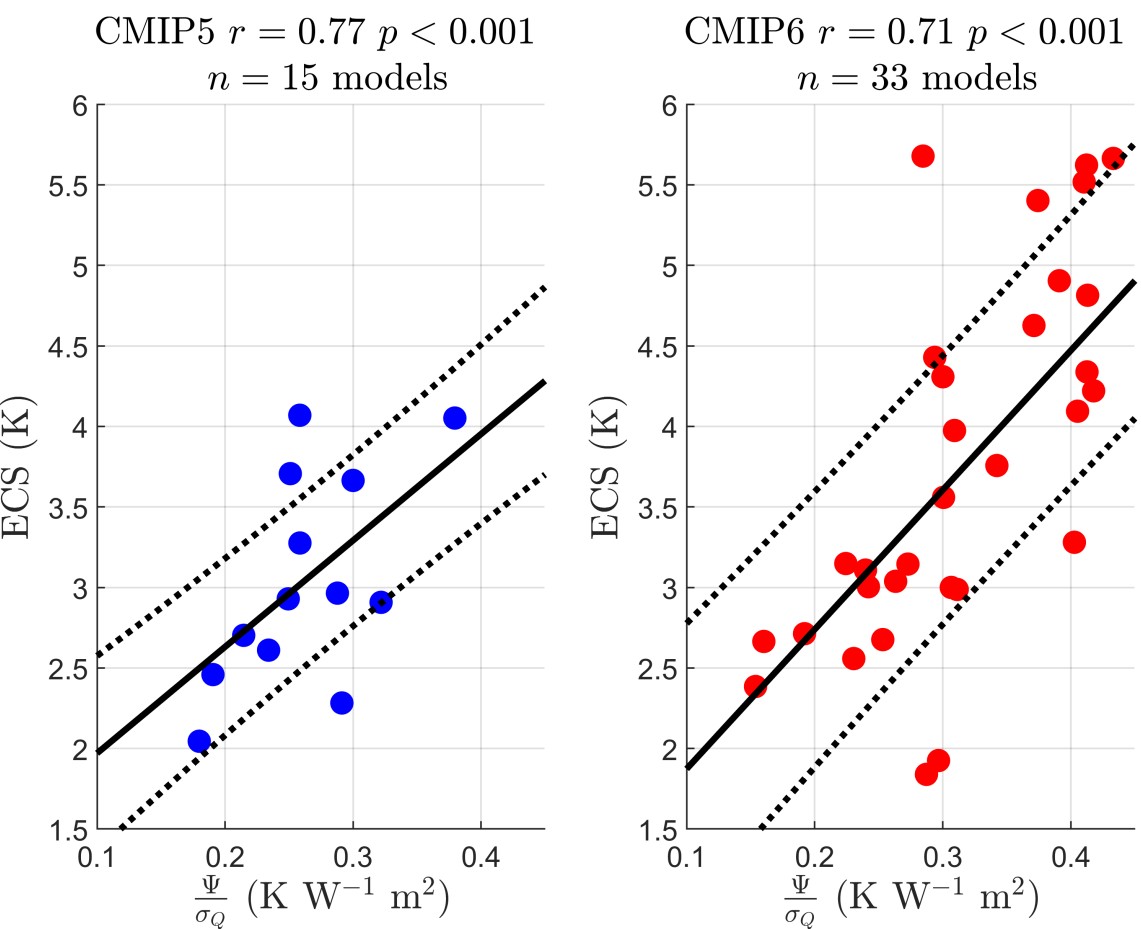

**Figure 3.** $\Psi/\sigma_Q$ against ECS in the CMIP5 (left panel) and CMIP6 (right panel) model ensembles running the historical experiment. The period 1880-2005 of each model's timeseries is used to calculate $\Psi$ and $\sigma_Q$. Individual models are plotted as circles (CMIP5 models are blue and CMIP6 models are red). The best fit line in the ordinary least squares sense is shown in black along with the standard deviation of the prediction error (black dotted line). Pearson correlation $r$ and $p$ value are given for each emergent relationship in each subplot title.

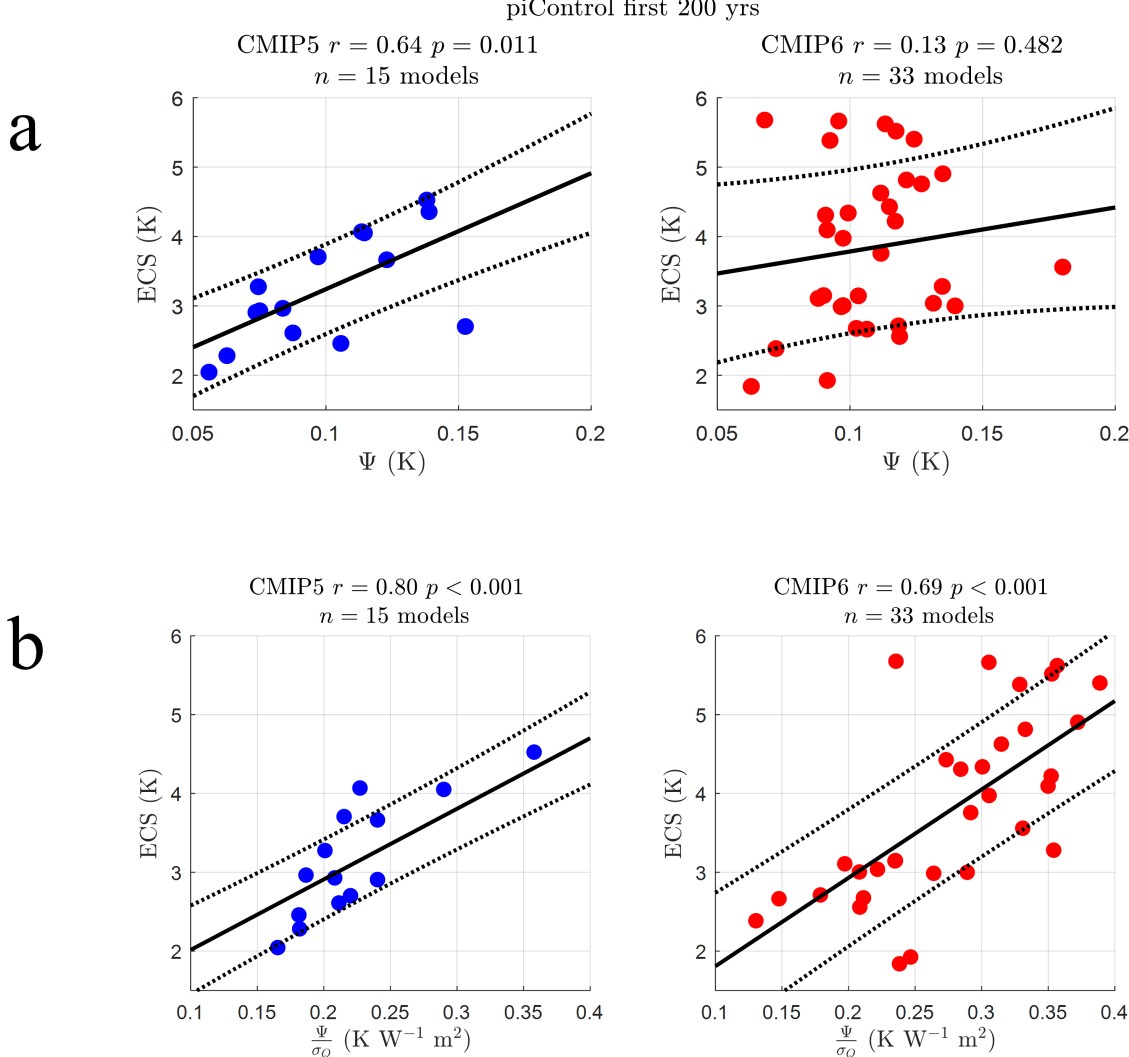

**Figure 4.** Individual models are plotted as circles (CMIP5 models are blue and CMIP6 models are red). The best fit line in the ordinary least squares sense is shown in black along with the standard deviation of the prediction error (black dotted line). Pearson correlation $r$ and $p$ value are given for each emergent relationship in each subplot title. The first 200 years of each model's timeseries is used to calculate $\Psi$ and $\sigma_Q$ running the piControl experiment (no external forcing). (a) $\Psi$ against ECS in the CMIP5 (left panel) and CMIP6 (right panel) model ensembles. (b) $\Psi/\sigma_Q$ against ECS in the CMIP5 (left panel) and CMIP6 (right panel) model ensembles.

H76 in equation 2 predicts $ECS = k\frac{Q_{2\times CO2}}{\sigma_Q}\Psi$ with a constant of proportionality $k = \sqrt{2}$. We investigate the empirical value of $k$ for the full complexity models next. As this relation should hold for all models running any experiment (provided you can remove the forced signal), we have plotted ECS against $\frac{Q_{2\times CO2}}{\sigma_Q}\Psi$ for an ensemble composed of all CMIP5 and CMIP6 models running both piControl and historical experiments (a total of 96 data points) to determine the empirical $k$ (figure 5 (a)). While the proportionality holds with a high correlation value and significance ($r = 0.74$, $p < 0.001$), the empirical constant is $k \sim 2\sqrt{2}$ rather than the $\sqrt{2}$ predicted by H76 i.e. the theoretical prediction of ECS from H76 is lower than the full complexity models suggest.

Williamson et al. (2018) showed the two-box and diffusion models also shared the linear ECS - $\Psi$ proportionality of H76 FAPP although with slightly different variables and constants. We have therefore also compared the more realistic two-box model theoretical predictions of $k$ (equation (23) in Williamson et al. (2018)) to the full complexity models (figure 5 (b)). Using the two-box values in tables B5 and B6 brings the empirically determined $k \sim 1.3\sqrt{2}$ closer to the two-box theoretical value ($k = \sqrt{2}$) with similar high correlation and significance ($r = 0.76$, $p < 0.001$). The two-box model adds a second, longer timescale to H76, mimicking the full complexity models more closely, however the theoretical $k$ is still slightly low. The lower prediction of $k$ than the empirical results suggest this could be due to the full complexity models having other timescales that the conceptual models do not. Although the conceptual models predict a linear ECS-$\Psi$ proportionality also seen in the full complexity CMIP models, they do not predict the constant of proportionality well. This is why the empirically determined $k$ should be used to obtain an emergent constraint as in CHW18.

## 5  Robustness to choice of model run

For both CMIP ensembles we have used one run for each model, preferably the one labelled or r1i1p1 (CMIP5) or r1i1p1f1 (CMIP6) where it exists, however we could have equally chosen any r*i*p* (CMIP5) or r*i*p*f* (CMIP6) for each model provided multiple runs of the same model exist. In this section we show the results for r1i1p1 or r1i1p1f1 are representative of a typical random run choice.

For models with multiple runs, we have drawn at random one run (r*i*p* or r*i*p*f* for CMIP5 and CMIP6 respectively) for each model and repeated the analysis in the previous sections multiple times. For the historical runs, in both CMIP5 and CMIP6 ensembles, many models do repeated runs, sometimes multiple times. For example, the CMIP6 model CanESM5 has the most runs, performing the historical experiment 50 times. There are $1.6 \times 10^{12}$ and $5.4 \times 10^{20}$ unique permutations for the same set of CMIP5 and CMIP6 models respectively performing the historical experiment. These numbers are clearly too large to search exhaustively. We have therefore drawn 1000 unique permutations for the historical experiment and repeated the analysis in this manuscript i.e. calculated the Pearson correlation, $r$, for every one of these 1000 permutations. The results are shown in the upper half of tables 1 (CMIP5) and 2 (CMIP6). We find that the results reported for r1i1p1 and r1i1p1f1 where they exist are fairly typical of a randomly chosen set of runs i.e. they fall within one standard deviation of the mean value in the CMIP5 ensemble. The CMIP6 historical experiment $r(\Psi, ECS)$ with r1i1p1f1 is slightly higher than would be expected (mean value for a randomly chosen permutation is $r = 0.18 \pm 0.11$, far from significant, compared to $r = 0.31$, almost

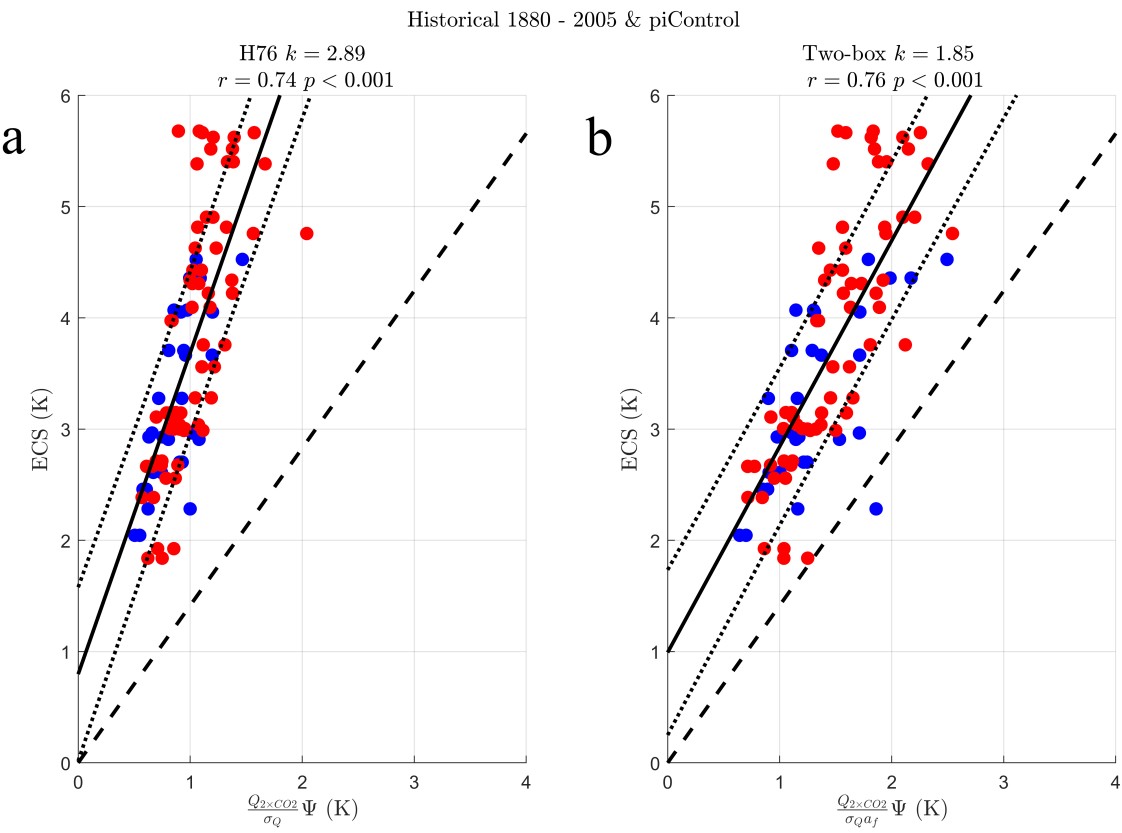

**Figure 5.** ECS against the theoretical predictor it is proportional to in (a) the one-box or H76 model and (b) the two-box model. Individual models are plotted as circles (CMIP5 models are blue and CMIP6 models are red) running both historical and piControl experiments between 1880-2005 and the first 200 years of the simulation respectively. The best fit line in the ordinary least squares sense is shown in black along with the standard deviation of the prediction error (black dotted line). Pearson correlation $r$ and $p$ value are given for each emergent relationship in each subplot title. The empirically determined constant of proportionality between the $x$ axis variable and ECS, $k$, is given in each subplot title. The H76 and two-box theoretical values of $k = \sqrt{2}$ are plotted as the dashed black line.

| CMIP5 experiment | Pearson correlation $r$ | r1i1p1 run | r*i*p* run | | |
|---|---|---|---|---|---|
| | | | mean $\pm$ std | min | max |
| historical | $r(\Psi, ECS)$ | 0.66 | $0.59 \pm 0.11$ | 0.14 | 0.82 |
| | $r(\sigma_Q, ECS)$ | -0.09 | $-0.16 \pm 0.08$ | $-0.40$ | 0.09 |
| | $r(\Psi/\sigma_Q, ECS)$ | 0.77 | $0.75 \pm 0.05$ | 0.58 | 0.89 |
| piControl | $r(\Psi, ECS)$ | 0.64 | $0.61 \pm 0.02$ | 0.57 | 0.64 |
| | $r(\sigma_Q, ECS)$ | -0.09 | $-0.10 \pm 0.02$ | $-0.13$ | $-0.08$ |
| | $r(\Psi/\sigma_Q, ECS)$ | 0.80 | $0.79 \pm 0.01$ | 0.77 | 0.80 |

**Table 1.** CMIP5 - Robustness of correlation to choice of run: Comparison of Pearson correlation $r$ for model ensemble using the r1i1p1 run (3rd column) with randomly chosen runs r*i*p*. There are 15 unique permutations of runs for the piControl experiment and $1.6 \times 10^{12}$ unique permutations for the historical experiment. We calculate $r$ for all unique permutations for the piControl experiment and 1000 randomly chosen unique permutations for the historical experiment. We report the mean value of $r$ with its standard deviation ($\pm$) in column 4 and the most extreme values of $r$ in columns 5 and 6.

significant). We have also listed the outer most values (min and max) found in the 1000 random run choice distribution for completeness. As one would expect with a large enough sample, there is a chance of finding non-representative correlations i.e. for the CMIP5 historical ensemble there is a small possibility that you might find an $r(\Psi, ECS) = 0.14$ (far from significant) or even $r(\Psi, ECS) = 0.82$ (highly significant) in a random pick of runs. These are not typical however.

For the piControl runs, there are 15 and 24 unique permutations for the same set of CMIP5 and CMIP6 models respectively. The relatively low number of unique permutations is due to the low number of repeated runs for the piControl experiment. We have run the same analysis in this manuscript i.e. calculated the Pearson correlation, $r$, for every one of these permutations. The results are shown in the lower half of tables 1 (CMIP5) and 2 (CMIP6). Because of the low number of unique permutations, the range of results is much narrower than the historical experiment. Again, we find that the results reported for r1i1p1 and

r1i1p1f1 where they exist are fairly typical of a randomly chosen set of runs i.e. they fall within one standard deviation of the mean value in the CMIP5 and CMIP6 ensembles with the exception of CMIP5 $r(\Psi, ECS) = 0.64$ (highly significant) for r1i1p1 compared to $r = 0.61 \pm 0.02$ (highly significant) for a randomly chosen permutation. This is actually highest value of $r$ (max) found in that experiment.

## 6    Can theory simulate the CMIP model results?

In section 4 we found that by including the forcing parameter $\sigma_Q$ in the predictor for ECS, an emergent relationship could be recovered for both CMIP5 and CMIP6 ensembles. These relationships are present in both the historical and piControl experiments giving confidence in the underlying theoretical basis FAPP.

    In this section we make a more demanding test of the theoretical basis by asking if theory alone can simulate the full complexity CMIP model ensemble $r(\Psi, ECS)$ and $r(\Psi/\sigma_Q, ECS)$ results. To do this, we create a H76 model emulator of

| CMIP6 experiment | Pearson correlation $r$ | r1i1p1f1 run | r*i*p*f* run | | |
|---|---|---|---|---|---|
| | | | mean $\pm$ std | min | max |
| historical | $r(\Psi, ECS)$ | 0.31 | $0.18 \pm 0.11$ | $-0.14$ | 0.46 |
| | $r(\sigma_Q, ECS)$ | -0.58 | $-0.58 \pm 0.04$ | $-0.69$ | $-0.46$ |
| | $r(\Psi/\sigma_Q, ECS)$ | 0.71 | $0.69 \pm 0.04$ | 0.56 | 0.77 |
| piControl | $r(\Psi, ECS)$ | 0.13 | $0.13 \pm 0.01$ | 0.11 | 0.15 |
| | $r(\sigma_Q, ECS)$ | -0.58 | $-0.57 \pm 0.01$ | $-0.58$ | $-0.55$ |
| | $r(\Psi/\sigma_Q, ECS)$ | 0.69 | $0.70 \pm 0.01$ | 0.69 | 0.71 |

**Table 2.** CMIP6 - Robustness of correlation to choice of run: Comparison of Pearson correlation $r$ for model ensemble using the r1i1p1f1 run if available (3rd column) with randomly chosen runs r*i*p*f*. There are 24 unique permutations of runs for the piControl experiment and $5.4 \times 10^{20}$ unique permutations for the historical experiment. We calculate $r$ for all unique permutations for the piControl experiment and 1000 randomly chosen unique permutations for the historical experiment. We report the mean value of $r$ with its standard deviation ($\pm$) in column 4 and the most extreme values of $r$ in columns 5 and 6.

each of the $i \in \{1, 2, ..., n\}$ full complexity CMIP5 and CMIP6 models used in the preceding figures. With the emulator H76 models we can build emulator H76 CMIP5 and CMIP6 ensembles and run analogous historical and piControl experiments with them. This will allow us to compare the results of the pure theory used in CHW18 with that of the full complexity CMIP model ensembles. We also fit the more complete two-box model in addition for comparison (see section A).

## 6.1 Methodology

The H76 model fitted to each of the full complexity CMIP models is given by equation 1. Parameters are fitted from the full complexity abrupt-4xCO2 CMIP model experiments: $\lambda$ and $Q_{2 \times CO2}$ are determined from Gregory plots (tables B1, B2), $C$ is found using a modification of Geoffroy et al. (2013b)'s methodology (tables B3 and B4).

Geoffroy et al. (2013b) published parameter values for two-box models fitted to CMIP5 models. The two-box model is H76's well mixed upper ocean/atmosphere box extended by coupling to a large heat capacity deep ocean box (see section A). This gives the two-box model a fast and a slow $e$-folding timescale of adjustment with typical values of $\sim 4$ and $\sim 200$ years when fitted to CMIP models (see Geoffroy et al. (2013b) and tables B5 and B6) which is known to do a good job reproducing the global annual mean temperature response of climate models.

As H76 only has one-box and therefore one timescale, it cannot capture both fast and slow responses of CMIP models. Because $\Psi$ is a statistic that is dominated by fast timescale processes of a few years, a feature H76 does capture well, we choose to fit the fast response with H76 by using Geoffroy et al. (2013b)'s fast timescale fitting methodology (see equation 18 in that paper). When modified for H76, this equation becomes

$$\tau = -\frac{t}{\log(1 - \frac{T(t)\lambda}{Q_{2 \times CO2}})}. \tag{5}$$

We fit H76's timescale parameter $\tau = \frac{C}{\lambda}$ (and therefore the heat capacity $C$) by averaging over the first 5 years of the abrupt-4xCO2 experiment. We choose the average over 5 years rather than the first 10 in Geoffroy et al. (2013b)'s two-box fits. This

is because the H76 fit gets worse as the number of years in the average increases (in the root mean square error of the fit). For the two-box fits we use Geoffroy et al. (2013b)'s methodology unmodified (see section A for complete details).

Fitted values of $\lambda$, $\tau$, $C$ and $\sigma_Q$ are reported for CMIP5 and CMIP6 ensembles in tables B3 and B4 respectively.

## 6.2 Emulator piControl experiments

We perform analogous piControl experiments with the H76 and two-box CMIP5 and CMIP6 ensembles by integrating each of

295 the individual CMIP H76 (and two-box) emulators (equations 1 and A1 respectively) numerically with forcing $Q_i(t)$, a zero mean random variable with model specific standard deviation $\sigma_{Qi}$ and Gaussian pdf. We write this as

$$Q_i(t) = \sigma_{Qi}\eta_i(t) \tag{6}$$

where $\eta_i(t)$ is the Gaussian random variable with unit standard deviation. The equations are integrated with a timestep of 0.1 yrs using the Euler-Maruyama method. The $T_i(t)$ timeseries that result are then analyzed in the same way as the full complexity

CMIP model timeseries to produce a pair of values $(\Psi_i, ECS_i)$. For the full set of $n$ two-box models in each CMIP emulator ensemble $r(\Psi, ECS)$ and $r(\Psi/\sigma_Q, ECS)$ are calculated. Because $Q_i(t)$ is a random variable of finite length, repeating the same experiment results in slightly different values of $r(\Psi/\sigma_Q, ECS)$ for each run due to the properties of statistical estimators (estimation converges as $1/\sqrt{N}$ where $N$ is the number of points in the timeseries). The same applies to different initial value runs in the full complexity models due to the chaotic weather variability the random forcing captures in the H76 and two-

box models. We therefore repeat each piControl emulator experiment 250 times and compare the distribution of emulator $r(\Psi, ECS)$ values with the single full-complexity CMIP piControl experiment.

Results are shown in figure 6. Agreement between the H76 emulator CMIP $r(\Psi, ECS)$ and the full complexity CMIP ensembles is reasonable. Full complexity CMIP5 ensemble $r(\Psi, ECS)$ results (LH panel, blue dotted line) fall in the upper end of the distribution of $r(\Psi, ECS)$ H76 emulator values. Although full complexity $r(\Psi, ECS)$ CMIP6 results (in red) were

310 shown to be lower in correlation (red dotted line), they can still be simulated reasonably well by the H76 emulator ensemble, falling like CMIP5, in upper end of simulated $r(\Psi, ECS)$ values. In the RH panel of figure 6 where $\Psi$ is now normalized by the mean amplitude of the random forcing $\sigma_Q$ both CMIP5 and CMIP6 results are much more similar, with histograms of the H76 emulator ensembles and the full complexity results having much more overlap although simulated values are still on average slightly lower than the full complexity ensembles.

Analogous figures simulated with two-box emulator CMIP ensembles are shown in figure A1. The two-box ensembles do an even better job of simulating the full complexity CMIP results.

## 6.3 Emulator historical experiments

The analogous historical experiments are performed in the same way to the piControl experiments but with a common external forcing component $Q_i(t)$ in addition to the random forcing. This comes from GHGs, volcanoes, solar cycles and others. For this

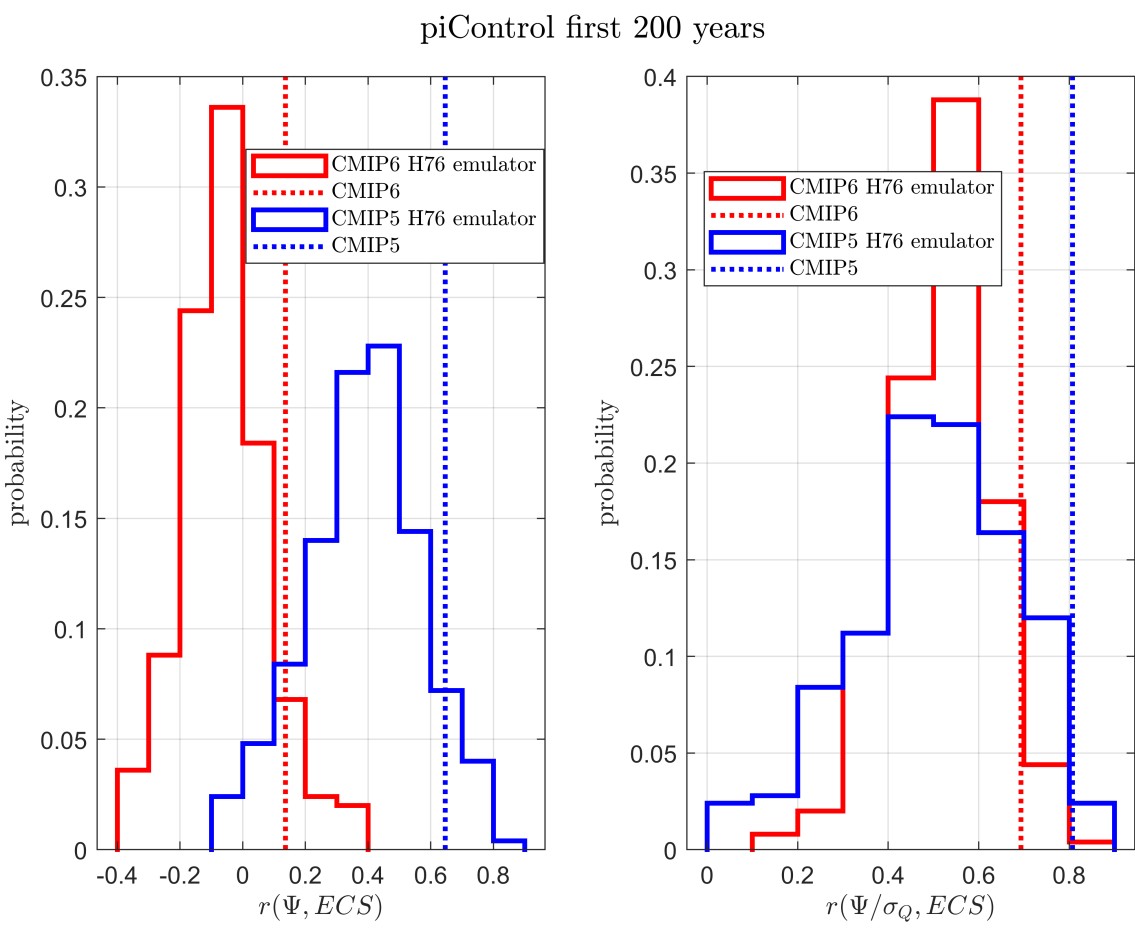

**Figure 6.** Probability of obtaining $r(\Psi, ECS)$ (left) and $r(\Psi/\sigma_Q, ECS)$ (right) in the H76 CMIP emulator ensembles performing a piControl simulation of 200 years. The CMIP5 emulator ensemble histogram is given in blue, equivalent CMIP6 in red. The full complexity CMIP ensemble results performing the same experiment are the vertical dotted lines.

common external forcing component we use Meinshausen et al. (2011) reconstructed historical forcing ($Q_{IPCC}(t)$). Explicitly

$$Q_i(t) = Q_{IPCC}(t) + \sigma_{Qi}\eta_i(t) \tag{7}$$

in the historical simulations. We integrate the H76 and two-box ensembles between the years 1765 and 2005 but calculate $\Psi$ and $r(\Psi, ECS)$ between 1880-2005 to correspond to the full complexity model analysis. Results are shown in figure 7. As with

325 the piControl experiments in section 6.2, agreement between the pure theory H76 emulator ensembles and the full complexity ensembles is reasonable, giving confidence that the underlying theory used in CHW18 is good FAPP. The analogous figure simulated with two-box CMIP emulator ensembles does an even better job (figure A2).

### 6.4    Emulator experiments with constant $\sigma_Q$

We have shown that by taking into account model specific $\sigma_Q$ in the CHW18 theory we can both understand $r(\Psi, ECS)$

correlation results and can recover good emergent relationships for both CMIP5 and CMIP6 in piControl and historical runs. In CHW18 and Cox et al. (2018b) it was assumed that $\sigma_Q$ was constant for each model in the CMIP5 ensemble (assumption **A3**). We now test this assumption with the H76 and two-box CMIP ensembles. Instead of fitting $\sigma_Q$ to each CMIP model we fix it to be a constant, $\sigma_Q = 0.25$ W m$^{-2}$ following Cox et al. (2018b). This value was chosen (even though it is lower than the values given in the tables) as it was the mean value of the standard deviation of net top-of-the-atmosphere radiation which

was thought to be a good proxy for $\sigma_Q$ at that time. Results with constant, model independent $\sigma_Q$ for $r(\Psi, ECS)$ are shown in figure 8 (H76) and A3 (two-box) for both piControl and historical experiments. $r(\Psi/\sigma_Q, ECS)$ results are not shown as they are identical to $r(\Psi, ECS)$. This is because the predictors, the set of $\{\Psi_i\}$ are all divided by the same constant.

The constant $\sigma_Q$ assumption can be seen to be good for the CMIP5 ensemble (blue) with full complexity models (blue dotted line) agreeing well with likely values of the H76 and two-box CMIP5 emulator ensembles (blue histogram). However,

the full complexity CMIP6 ensemble (red dotted line) correlations are generally much lower than the CMIP6 emulators (red histogram). This is again supporting evidence that the underlying theory in CHW18 is sound FAPP. The similarity in the CMIP5 and CMIP6 histograms also suggests there is no real difference in the parameters of the emulator ensembles. The difference can be attributed to the amount of correlation between $\sigma_Q$ and ECS in the CMIP5 and CMIP6 ensembles.

### 7    Discussion and conclusion

The aim of this manuscript was to understand why the strong emergent relationship from CHW18 found in the CMIP5 model ensemble weakened in the newer CMIP6 ensemble. This emergent relationship was based on reasonable, although simple physical principles so it is interesting (and important) to understand the differences between the theory and full complexity models. A number of assumptions (section 3) were made in deriving the theoretical emergent relationship between the predictor $\Psi$, a metric based on annual global mean temperature variability and ECS, the predictand in CHW18. We have shown the 'no

correlation between forcing and ECS' assumption no longer holds for the CMIP6 ensemble. In particular, the parameter $\sigma_Q$

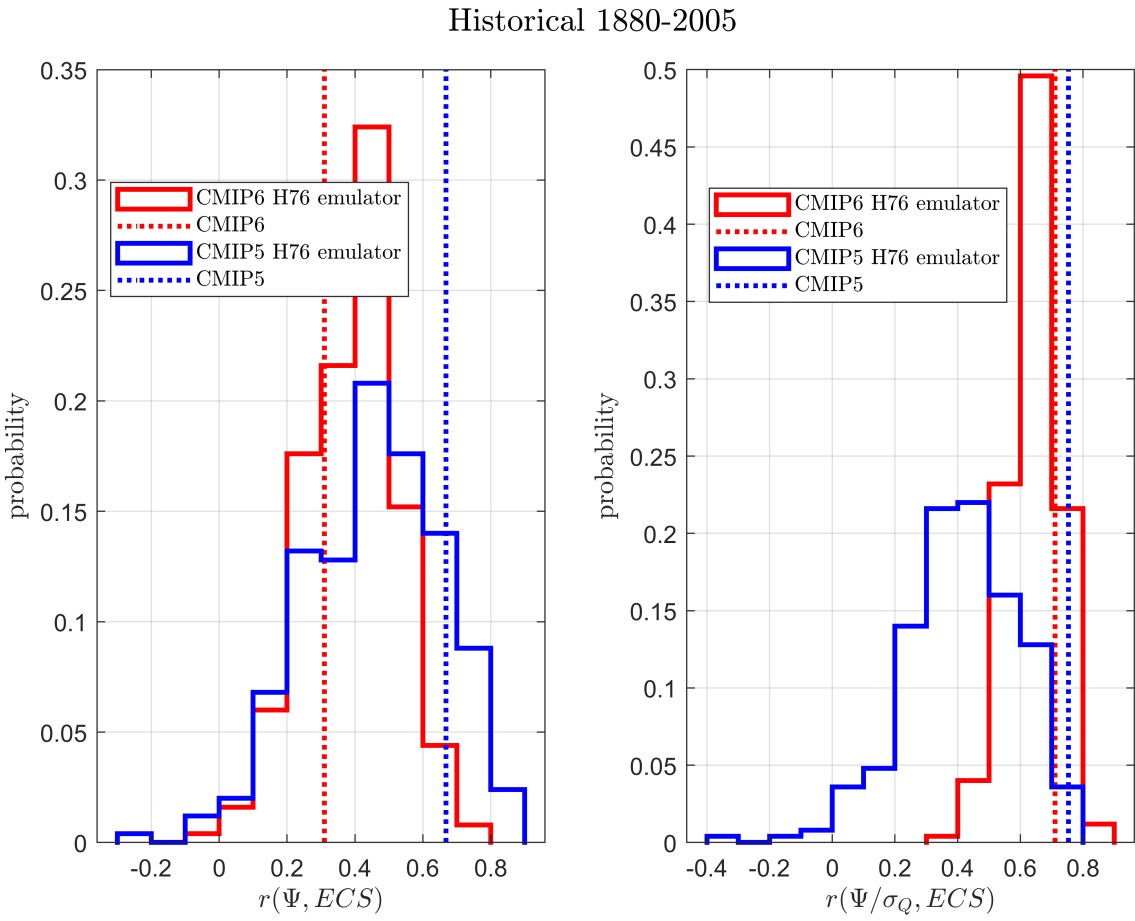

**Figure 7.** Probability of obtaining $r(\Psi, ECS)$ (left) and $r(\Psi/\sigma_Q, ECS)$ (right) in the H76 CMIP emulator ensembles performing a historical simulation of the period 1880-2005. The CMIP5 emulator ensemble histogram is given in blue, equivalent CMIP6 in red. The full complexity CMIP ensemble results performing the same experiment are the vertical dotted lines.

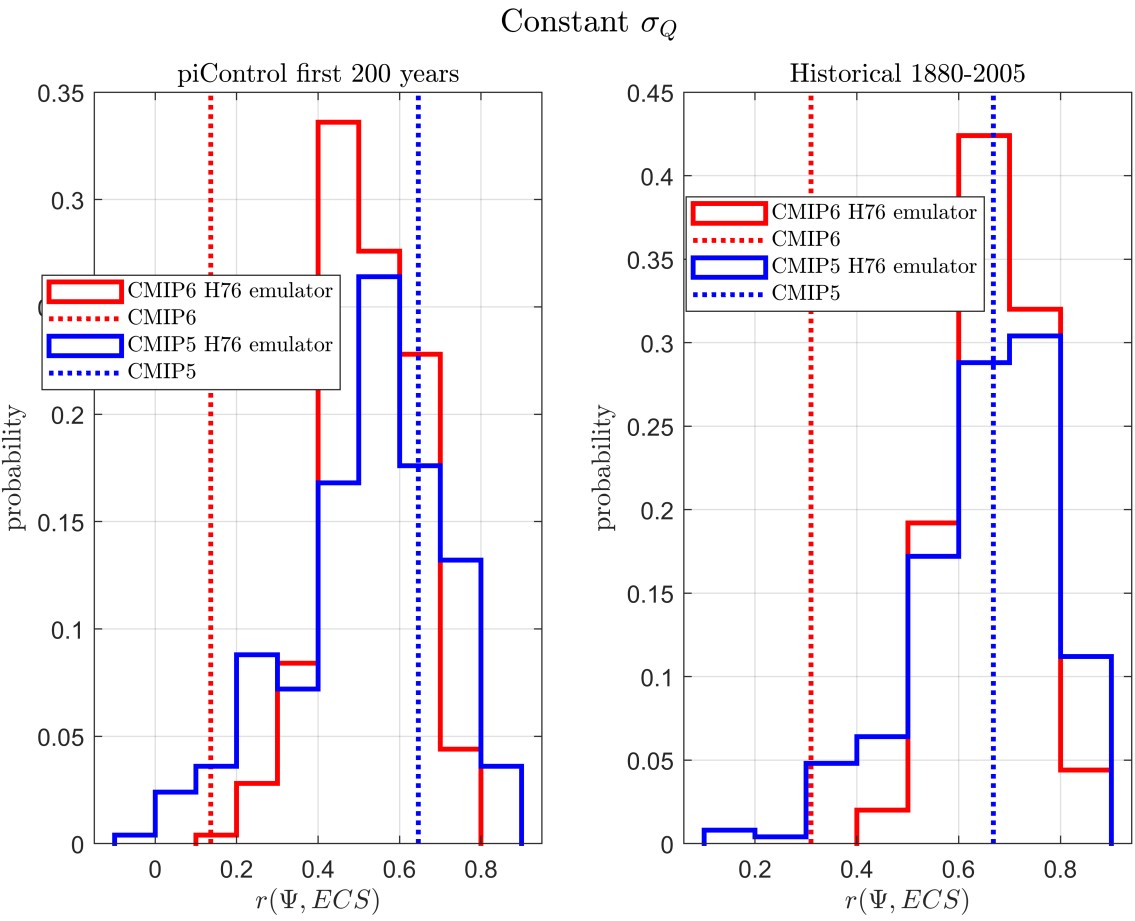

**Figure 8.** Probability of obtaining $r(\Psi, ECS)$ in the H76 CMIP emulator ensembles performing a piControl (left panel) and historical (right panel) simulation if each two-box emulator is given the same value of $\sigma_Q = 0.25$ W m$^{-2}$. The CMIP5 emulator ensemble histogram is given in blue, equivalent CMIP6 in red. The full complexity CMIP ensemble results performing the same experiment are the vertical dotted lines.

describing random forcing from internally generated variability, *is* correlated to ECS in CMIP6 and when this parameter is incorporated into the predictand, a good emergent relationship is recovered for both CMIP ensembles.

Assumption **A3** stated that the forcing parameters, $Q_{2\times CO2}$ and $\sigma_Q$ could be treated as constants across a model ensemble. While this is a fair assumption for $Q_{2\times CO2}$ for both CMIP ensembles and $\sigma_Q$ in the CMIP5 ensemble, we have shown that $\sigma_Q$ is correlated to ECS in the CMIP6 ensemble. We have also shown that when the predictor of ECS is changed to $\Psi/\sigma_Q$ good emergent relationships are recovered in both CMIP ensembles for both piControl and historical experiments. We also showed that pure theory could reproduce the full-complexity CMIP model results using H76 and two-box CMIP emulator ensembles. Although the proportionality between ECS and the predictor $\frac{Q_{2\times CO2}}{\sigma_Q}\Psi$ has a high correlation and significance, simple pure theory underestimates the constant of proportionality. Aside from this, these results give us confidence the theoretical basis of CHW18 still applies as well to CMIP6 models as it did for CMIP5 FAPP. Testing the theoretical basis was the underlying aim of our study.

Several questions remain however: Can we estimate $\sigma_Q$ from observations and therefore get an emergent constraint on ECS from the CMIP6 ensemble? Why is $\sigma_Q$ correlated to ECS in CMIP6 and not CMIP5? $\sigma_Q$ is a parameter designed to reproduce the observed global annual mean temperature variability, $\sigma_T$, in the non-chaotic H76 and two-box models. In the full complexity models and the real world, this parameter attempts to capture chaotic internal variability as well as sub-annual (fast) feedbacks. It is fitted in this study using $\sigma_T$ (an observable) as well as the unobservable two-box parameters. The reliance on these unobservable two-box parameters makes it appear that getting an estimate of $\sigma_Q$ in the real world and so an emergent constraint, may be tough. However there may be observable proxies for $\sigma_Q$ which we have not yet found.

An obvious place to start looking for a proxy for $\sigma_Q$ is in basic theory. The simplest one can imagine

$$Q(t) = N(t) + \lambda T(t) \tag{8}$$

where $N(t)$ is the net top-of-the-atmosphere radiative flux. However this still requires knowledge of $\lambda$. Even given knowledge of $\lambda$ it is well known (Forster, 2016) that $N$ is poorly correlated to $T$ where most of the change in $N$ and $T$ is driven by internal variability (although this relation works very well for large forced trends, for example the Gregory method works well applied to large stepped increases in $CO_2$). There are several models (Winton et al., 2010; Geoffroy et al., 2013a) and methods (Dessler et al., 2018; Bloch-Johnson et al., 2020) that get much better correlations between $N$ and $T$ when most of the changes are driven internally by taking into account the spatial distributions (the so called pattern effect, see Armour et al. (2012)). We leave this to a future study.

The question of why $\sigma_Q$ is correlated to ECS in CMIP6 and not CMIP5 is also left unanswered. However, one can speculate why this may be the case: As previously mentioned $\sigma_Q$ is a fitting parameter that is designed to capture the effect of chaotic internal variability as well as sub-annual (fast) feedbacks on global mean temperature variability. Zelinka et al. (2020) showed that the increased range of ECS in the CMIP6 models could be explained by the increased range in cloud feedbacks (see also Bock and Lauer (2024)). As $\sigma_Q$ is fitted to annual temperature timeseries, some of this fast (sub-annual) cloud feedback effect could be included in $\sigma_Q$ correlating it to ECS. We leave concrete answers to a future study.

We have understood what assumption in the theoretical emergent relationship for CHW18 was responsible for the weakened correlation in CMIP6, namely $\sigma_Q$ is correlated with ECS. When accounted for, good emergent relationships are recovered. Although the information that the simple theory holds FAPP is useful and that $\sigma_Q$ is correlated in CMIP6 with ECS is interesting, it is disappointingly not useful in constraining ECS due to the unobservable nature (we think) of $\sigma_Q$. In this sense, the method in CHW18 does not produce a useful emergent constraint on CMIP6 because the extra degree of freedom in $\sigma_Q$ needs to be incorporated.

Schlund et al. (2020) tested 11 emergent constraints found in CMIP5 and nearly all of these got weaker in CMIP6. We do not know whether they failed for similar reasons. Indeed, many of them do not have a simple theoretical model as a basis for their emergent relationship so assumption testing, the approach we follow in this manuscript, would be difficult to do. This is why we argue that emergent constraints should be based on a testable, falsifiable theoretical model. This aids understanding and lifts emergent constraint research from looking for strong correlations between variables to a more scientific approach of testing hypotheses of how the Earth system works. However, looking at all these other emergent constraints and identifying why they got weaker in CMIP6 would be very beneficial to understanding and useful to the community.

Emergent constraints based on theory with minimal degrees of freedom are most likely to be the most robust and useful. Constraints such as Hall and Qu (2006) on snow albedo feedback where the predictor (seasonal cycle snow albedo feedback) and predictand (climate change snow albedo feedback) are the essentially the same variable have been shown to be robust through 3 CMIP generations (Thackeray et al., 2021). Other constraints of this type that are likely to be more robust are the transient climate response constraints of Nijsse et al. (2020) and Tokarska et al. (2020) where near term historical warming is the predictor of future, longer term warming.

Even if emergent relationships based on sound theoretical principles do fail there is still information to be gleaned on understanding why. Today, there is even more of an opportunity for the top down insights of specific conceptual models to meet and complement the comprehensive, bottom up approach from state-of-the-art climate models; there are many more high quality observations; the global warming signal has also become clearer over time; and there is also a large archive of past and present climate model simulations.

*Data availability.* All original CMIP5 and CMIP6 data used in this study are publicly available at https://esgf-node.llnl.gov/projects/cmip5/and https://esgf-node.llnl.gov/projects/cmip6/ respectively (last access: August 2021).

**Appendix A: Two-box CMIP emulators**

The two-box model is H76's low thermal inertia atmosphere/well-mixed ocean surface layer with heat capacity $C$ extended with a large heat capacity $C_0$ deep ocean box coupled to the surface box by flux $\gamma$. This gives the model two timescales of adjustment, a fast ($\tau_f$) and a slow $e$-folding time ($\tau_s$). When fitted to CMIP models typical values for the timescales are $\tau_f \sim 4$ years and $\tau_s \sim 200$ years (see tables B5 and B6).

Each CMIP model labelled with $i$ is 'mimicked' by the two-box equations

$$C_i \frac{dT_i}{dt} = Q_i(t) - \lambda_i T_i(t) - \gamma_i \left( T_i(t) - T_{0i}(t) \right),$$ (A1)

$$C_{0i} \frac{dT_{0i}}{dt} = \gamma_i \left( T_i(t) - T_{0i}(t) \right).$$

$T_{0i}$ is the annual global mean deep ocean temperature anomaly of model $i$. Parameters are fitted from the full complexity abrupt-4xCO2 CMIP model experiments. The parameters $\lambda$ and $Q_{2 \times CO2}$ are determined from Gregory plots while $C$, $C_0$ and $\gamma$ are determined using Geoffroy's methodology (Geoffroy et al., 2013b). We use Geoffroy et al. (2013b)'s published values for CMIP5 models. Values for CMIP6 models are given in tables B2 and B6.

The standard deviation of white noise forcing $\sigma_Q$ is fitted for each model from the global annual mean temperature timeseries of either the piControl or historical experiment. This timeseries is linearly detrended with a rolling 55 year window. This is to isolate the $T(t)$ response to internal variability, analogous to how $\Psi$ is determined in the CHW18 methodology, to leave the noisy $T(t)$ response to white noise with standard deviation $\sigma_T$. The theoretical formula is given by Williamson et al. (2018)

$$\sigma_T^2 = \frac{\sigma_Q^2}{2\lambda^2} \left( \frac{a_f^2}{\tau_f} + \frac{a_s^2}{\tau_s} + \frac{4 a_f a_s}{\tau_f + \tau_s} \right)$$ (A2)

We rearrange this relation to get $\sigma_Q$. Values of $\lambda$, $a_f$, $a_s$, $\tau_f$ and $\tau_s$ are taken from tables B1, B2, B5 and B6. Values of $\sigma_Q$ in both historical and piControl runs are also reported in tables B5 and B6. The parameters $a_f$, $a_s$, $\tau_f$ and $\tau_s$ are complicated functions of the parameters $C$, $C_0$, $\lambda$ and $\gamma$. Their exact full functional forms can be found in Geoffroy et al. (2013b) and are not given here.

## Appendix B: Parameter values

*Author contributions.* M.S.W. carried out the data analysis and drafted the paper with advice from P.M.C., C.H. and F.J.M.M.N. All authors contributed to the submitted paper.

*Competing interests.* The contact author has declared that neither they nor their co-authors has any competing interests.

*Acknowledgements.* This work was supported by the European Research Council (ERC) ECCLES project, grant agreement number 742472 (M.S.W., P.M.C. and F.J.M.M.N.); the EU Horizon 2020 Research Programme CRESCENDO project, grant agreement number 641816 (M.S.W. and P.M.C.); the Horizon Europe project OptimESM, grant agreement number 101081193 (P.M.C.); and the NERC CEH National Capability fund (C.H.). We also acknowledge the World Climate Research Programme's Working Group on Coupled Modelling, which is responsible for CMIP, and we thank the climate modelling groups for producing and making available their model output.

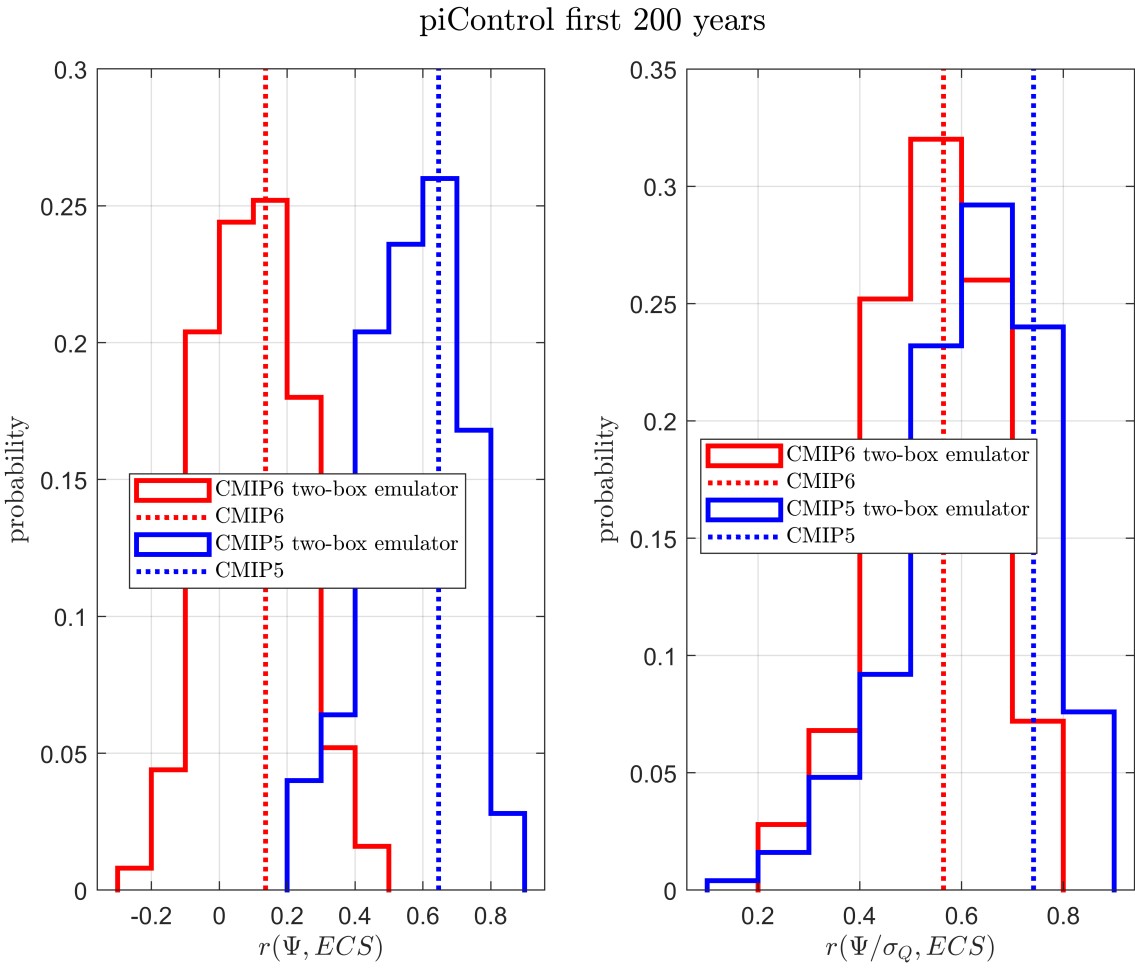

**Figure A1.** Probability of obtaining $r(\Psi, ECS)$ (left) and $r(\Psi/\sigma_Q, ECS)$ (right) in the two-box CMIP emulator ensembles performing a piControl simulation of 200 years. The CMIP5 emulator ensemble histogram is given in blue, equivalent CMIP6 in red. The full complexity CMIP ensemble results performing the same experiment are the vertical dotted lines.

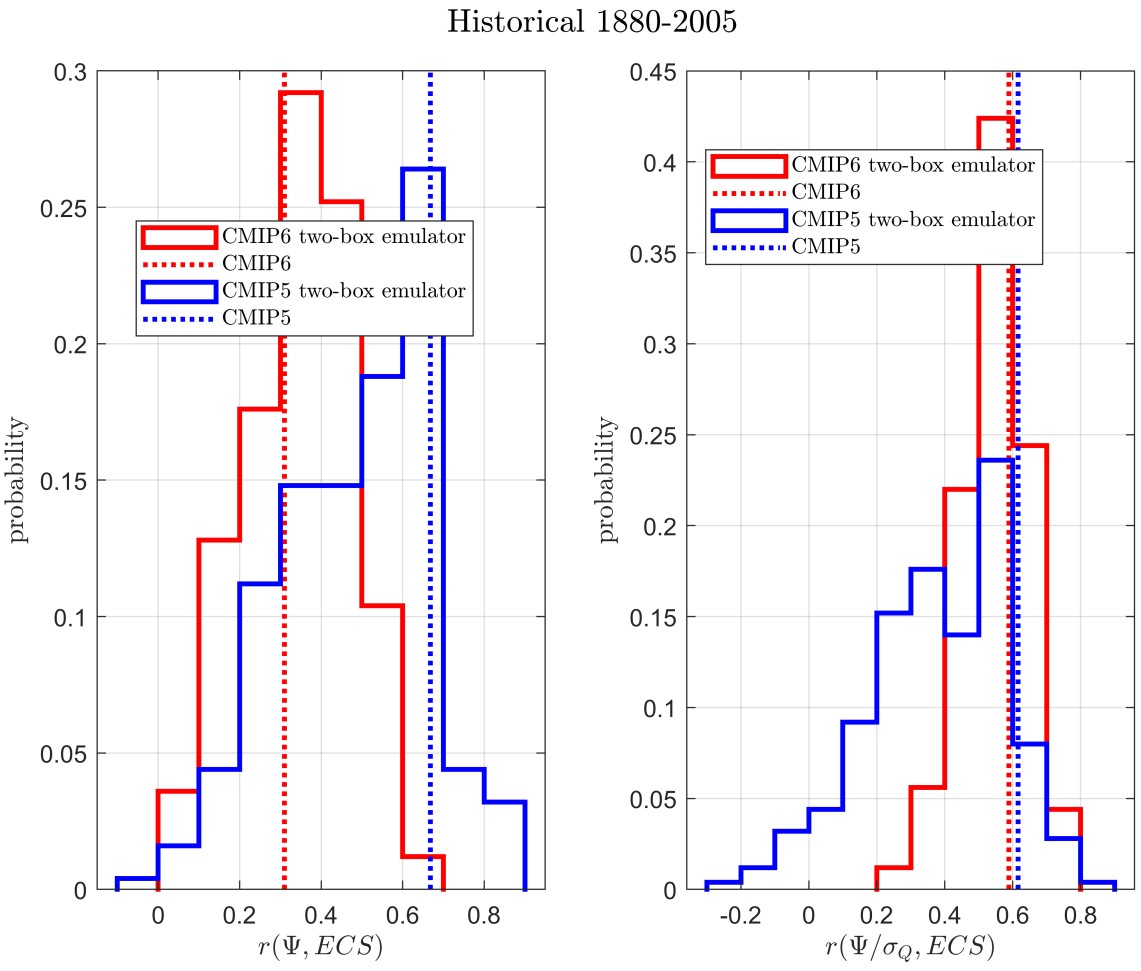

**Figure A2.** Probability of obtaining $r(\Psi, ECS)$ (left) and $r(\Psi/\sigma_Q, ECS)$ (right) in the two-box CMIP emulator ensembles performing a historical simulation of the period 1880-2005. The CMIP5 emulator ensemble histogram is given in blue, equivalent CMIP6 in red. The full complexity CMIP ensemble results performing the same experiment are the vertical dotted lines.

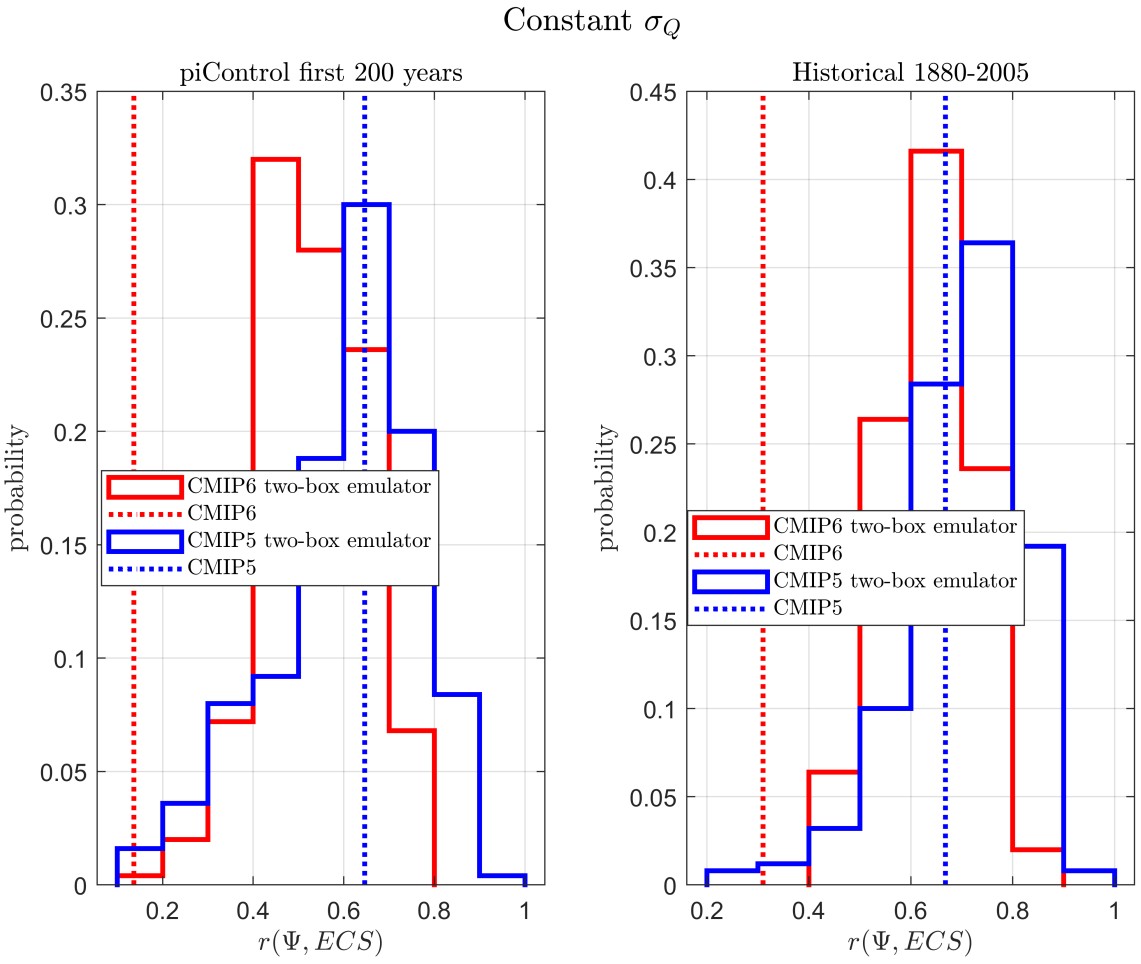

**Figure A3.** Probability of obtaining $r(\Psi, ECS)$ in the two-box CMIP emulator ensembles performing a piControl (left panel) and historical (right panel) simulation if each two-box emulator is given the same value of $\sigma_Q = 0.25$ W m$^{-2}$. The CMIP5 emulator ensemble histogram is given in blue, equivalent CMIP6 in red. The full complexity CMIP ensemble results performing the same experiment are the vertical dotted lines.

| Model | $Q_{2\times CO2}$ (W m$^{-2}$) | $\lambda$ (W m$^{-2}$ K$^{-1}$) | ECS (K) |
|---|---|---|---|
| BNU-ESM | 3.70 | 0.93 | 4.00 |
| CCSM4 | 3.60 | 1.24 | 2.90 |
| CNRM-CM5 | 3.65 | 1.11 | 3.25 |
| CSIRO-Mk3-6-0 | 2.55 | 0.61 | 4.15 |
| CanESM2 | 3.80 | 1.03 | 3.70 |
| GFDL-ESM2M | 3.30 | 1.34 | 2.45 |
| GISS-E2-R | 3.65 | 1.70 | 2.15 |
| HadGEM2-ES | 2.95 | 0.65 | 4.55 |
| IPSL-CM5A-LR | 3.20 | 0.79 | 4.05 |
| MIROC5 | 4.25 | 1.58 | 2.70 |
| MPI-ESM-LR | 4.10 | 1.14 | 3.65 |
| MRI-CGCM3 | 3.30 | 1.26 | 2.60 |
| NorESM1-M | 3.10 | 1.11 | 2.80 |
| bcc-csm1-1 | 3.35 | 1.21 | 2.80 |
| inmcm4 | 3.10 | 1.51 | 2.05 |
| Multimodel mean | 3.44 | 1.15 | 3.19 |
| Standard deviation | 0.44 | 0.32 | 0.78 |

**Table B1.** Gregory plot determined parameters for CMIP5 models from Geoffroy et al. (2013b).

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

|  | $Q_{2\times CO2}$ | $\lambda$ | ECS |
| Model | (W m$^{-2}$) | (W m$^{-2}$ K$^{-1}$) | (K) |
| --- | --- | --- | --- |
| ACCESS-CM2 | 3.21 | 0.67 | 4.81 |
| ACCESS-ESM1-5 | 2.71 | 0.68 | 3.97 |
| AWI-CM-1-1-MR | 3.71 | 1.18 | 3.15 |
| BCC-CSM2-MR | 2.95 | 0.98 | 3.00 |
| BCC-ESM1 | 2.95 | 0.90 | 3.28 |
| CESM2-WACCM | 3.08 | 0.63 | 4.90 |
| CIESM | 3.80 | 0.67 | 5.68 |
| CMCC-CM2-SR5 | 3.67 | 1.03 | 3.56 |
| CanESM5 | 3.63 | 0.64 | 5.66 |
| E3SM-1-0 | 3.23 | 0.60 | 5.38 |
| EC-Earth3 | 3.30 | 0.78 | 4.22 |
| EC-Earth3-Veg | 3.32 | 0.77 | 4.34 |
| FIO-ESM-2-0 | 3.59 | 0.83 | 4.31 |
| GFDL-CM4 | 2.91 | 0.71 | 4.09 |
| GFDL-ESM4 | 3.51 | 1.31 | 2.68 |
| GISS-E2-1-G | 3.89 | 1.43 | 2.71 |
| GISS-E2-1-H | 3.54 | 1.14 | 3.11 |
| HadGEM3-GC31-LL | 3.38 | 0.60 | 5.62 |
| HadGEM3-GC31-MM | 3.36 | 0.61 | 5.52 |
| IITM-ESM | 4.37 | 1.83 | 2.38 |
| INM-CM4-8 | 2.61 | 1.42 | 1.84 |
| INM-CM5-0 | 2.88 | 1.49 | 1.93 |
| IPSL-CM6A-LR | 3.32 | 0.72 | 4.63 |
| MIROC-ES2L | 4.13 | 1.55 | 2.66 |
| MIROC6 | 3.76 | 1.47 | 2.56 |
| MPI-ESM-1-2-HAM | 3.93 | 1.31 | 3.00 |
| MPI-ESM1-2-HR | 3.58 | 1.20 | 2.99 |
| MPI-ESM1-2-LR | 4.08 | 1.34 | 3.04 |
| MRI-ESM2-0 | 3.36 | 1.07 | 3.14 |
| NESM3 | 3.73 | 0.78 | 4.76 |
| SAM0-UNICON | 3.83 | 1.02 | 3.76 |
| TaiESM1 | 3.75 | 0.85 | 4.43 |
| UKESM1-0-LL | 3.56 | 0.66 | 5.40 |
| Multimodel mean | 3.47 | 1.00 | 3.83 |
| Standard deviation | 0.42 | 0.34 | 1.15 |

Table B2. Gregory plot determined parameters for CMIP6 models.

| Model | $C$ (W yr m$^{-2}$ K$^{-1}$) | $\tau$ (yr) | historical $\sigma_Q$ (W m$^{-2}$) | piControl $\sigma_Q$ (W m$^{-2}$) |
|---|---|---|---|---|
| BNU-ESM | 7.9 | 8.5 | 0.53 | 0.50 |
| CCSM4 | 8.8 | 7.3 | 0.61 | 0.46 |
| CNRM-CM5 | 8.6 | 7.8 | 0.57 | 0.43 |
| CSIRO-Mk3-6-0 | 7.2 | 13.2 | 0.40 | 0.35 |
| CanESM2 | 8.0 | 7.9 | 0.60 | 0.46 |
| GFDL-ESM2M | 9.5 | 7.3 | 0.75 | 0.61 |
| GISS-E2-R | 8.1 | 5.4 | 0.34 | 0.40 |
| HadGEM2-ES | 9.2 | 14.2 | 0.51 | 0.39 |
| IPSL-CM5A-LR | 8.2 | 10.5 | 0.52 | 0.38 |
| MIROC5 | 9.6 | 6.1 | 0.95 | 0.60 |
| MPI-ESM-LR | 8.1 | 7.5 | 0.47 | 0.50 |
| MRI-CGCM3 | 8.9 | 7.3 | 0.39 | 0.40 |
| NorESM1-M | 9.3 | 9.0 | 0.51 | 0.41 |
| bcc-csm1-1 | 8.6 | 7.5 | 0.55 | 0.38 |
| inmcm4 | 9.6 | 6.4 | 0.39 | 0.33 |
| Multimodel mean | 8.6 | 8.4 | 0.54 | 0.44 |
| Standard deviation | 0.7 | 2.5 | 0.16 | 0.08 |

**Table B3.** H76 model parameters fitted from abrupt4xCO2 runs for CMIP5 models. $\sigma_Q$ values are calculated from detrended $T(t)$ for either historical or piControl runs.

| Model | $C$ (W yr m$^{-2}$ K$^{-1}$) | $\tau$ (yr) | historical $\sigma_Q$ (W m$^{-2}$) | piControl $\sigma_Q$ (W m$^{-2}$) |
|---|---|---|---|---|
| ACCESS-CM2 | 7.9 | 11.9 | 0.48 | 0.32 |
| ACCESS-ESM1-5 | 7.9 | 11.6 | 0.35 | 0.33 |
| AWI-CM-1-1-MR | 7.5 | 6.4 | 0.48 | 0.39 |
| BCC-CSM2-MR | 8.7 | 8.8 | 0.51 | 0.45 |
| BCC-ESM1 | 8.0 | 8.9 | 0.43 | 0.34 |
| CESM2-WACCM | 8.2 | 13.1 | 0.45 | 0.36 |
| CIESM | 8.6 | 12.8 | 0.33 | 0.30 |
| CMCC-CM2-SR5 | 8.0 | 7.7 | 0.57 | 0.64 |
| CanESM5 | 7.5 | 11.8 | 0.41 | 0.31 |
| E3SM-1-0 | 7.4 | 12.4 | 0.40 | 0.30 |
| EC-Earth3 | 7.2 | 9.3 | 0.53 | 0.33 |
| EC-Earth3-Veg | 7.1 | 9.2 | 0.46 | 0.34 |
| FIO-ESM-2-0 | 8.5 | 10.2 | 0.43 | 0.32 |
| GFDL-CM4 | 6.3 | 8.9 | 0.38 | 0.28 |
| GFDL-ESM4 | 8.3 | 6.3 | 0.70 | 0.50 |
| GISS-E2-1-G | 8.5 | 5.9 | 0.84 | 0.67 |
| GISS-E2-1-H | 8.6 | 7.6 | 0.52 | 0.45 |
| HadGEM3-GC31-LL | 8.0 | 13.4 | 0.40 | 0.32 |
| HadGEM3-GC31-MM | 8.1 | 13.3 | 0.37 | 0.32 |
| IITM-ESM | 10.2 | 5.5 | 0.79 | 0.55 |
| INM-CM4-8 | 7.3 | 5.1 | 0.40 | 0.30 |
| INM-CM5-0 | 8.0 | 5.3 | 0.43 | 0.35 |
| IPSL-CM6A-LR | 6.6 | 9.2 | 0.36 | 0.34 |
| MIROC-ES2L | 10.8 | 7.0 | 0.93 | 0.73 |
| MIROC6 | 8.7 | 5.9 | 0.72 | 0.61 |
| MPI-ESM-1-2-HAM | 8.6 | 6.6 | 0.55 | 0.55 |
| MPI-ESM1-2-HR | 7.9 | 6.6 | 0.50 | 0.43 |
| MPI-ESM1-2-LR | 8.7 | 6.5 | 0.68 | 0.52 |
| MRI-ESM2-0 | 9.2 | 8.6 | 0.54 | 0.42 |
| NESM3 | 5.3 | 6.7 | 0.42 | 0.29 |
| SAM0-UNICON | 8.4 | 8.3 | 0.51 | 0.40 |
| TaiESM1 | 8.0 | 9.5 | 0.45 | 0.38 |
| UKESM1-0-LL | 7.2 | 10.9 | 0.47 | 0.33 |
| Multimodel mean | 8.0 | 8.8 | 0.51 | 0.41 |
| Standard deviation | 1.0 | 2.6 | 0.15 | 0.12 |

**Table B4.** H76 model parameters fitted from abrupt4xCO2 runs for CMIP6 models. $\sigma_Q$ values are calculated from detrended $T(t)$ for either historical or piControl runs.

| Model | $C$ | $C_0$ | $\gamma$ | $\tau_f$ | $\tau_s$ | $a_f$ | $a_s$ | historical $\sigma_Q$ | piControl $\sigma_Q$ |
| --- | --- | --- | --- | --- | --- | --- | --- | --- | --- |
| | (W yr m$^{-2}$ K$^{-1}$) | (W yr m$^{-2}$ K$^{-1}$) | (W m$^{-2}$ K$^{-1}$) | (yr) | (yr) | | | (W m$^{-2}$) | (W m$^{-2}$) |
| BNU-ESM | 7.4 | 90 | 0.53 | 5.0 | 267 | 0.62 | 0.38 | 0.64 | 0.60 |
| CCSM4 | 6.1 | 69 | 0.93 | 2.8 | 132 | 0.56 | 0.44 | 0.65 | 0.48 |
| CNRM-CM5 | 8.4 | 99 | 0.50 | 5.2 | 289 | 0.68 | 0.32 | 0.67 | 0.50 |
| CSIRO-Mk3-6-0 | 6.0 | 69 | 0.88 | 3.9 | 200 | 0.38 | 0.62 | 0.52 | 0.46 |
| CanESM2 | 7.3 | 71 | 0.59 | 4.5 | 193 | 0.63 | 0.37 | 0.70 | 0.54 |
| GFDL-ESM2M | 8.1 | 105 | 0.90 | 3.6 | 197 | 0.59 | 0.41 | 0.87 | 0.71 |
| GISS-E2-R | 4.7 | 126 | 1.16 | 1.6 | 184 | 0.58 | 0.42 | 0.32 | 0.37 |
| HadGEM2-ES | 6.5 | 82 | 0.55 | 5.3 | 280 | 0.52 | 0.48 | 0.58 | 0.44 |
| IPSL-CM5A-LR | 7.7 | 95 | 0.59 | 5.5 | 286 | 0.56 | 0.44 | 0.65 | 0.47 |
| MIROC5 | 8.3 | 145 | 0.76 | 3.5 | 285 | 0.66 | 0.34 | 1.07 | 0.67 |
| MPI-ESM-LR | 7.3 | 71 | 0.72 | 3.9 | 164 | 0.60 | 0.40 | 0.55 | 0.59 |
| MRI-CGCM3 | 8.5 | 64 | 0.66 | 4.3 | 150 | 0.63 | 0.37 | 0.46 | 0.47 |
| NorESM1-M | 8.0 | 105 | 0.88 | 4.0 | 218 | 0.55 | 0.45 | 0.60 | 0.48 |
| bcc-csm1-1 | 7.6 | 53 | 0.67 | 4.0 | 126 | 0.62 | 0.38 | 0.62 | 0.43 |
| inmcm4 | 8.6 | 317 | 0.65 | 4.0 | 698 | 0.70 | 0.30 | 0.43 | 0.37 |
| Multimodel mean | 7.4 | 104 | 0.73 | 4.1 | 245 | 0.59 | 0.41 | 0.62 | 0.51 |
| Standard deviation | 1.1 | 64 | 0.19 | 1.0 | 138 | 0.08 | 0.08 | 0.18 | 0.10 |

**Table B5.** Two-box parameters fitted from abrupt4xCO2 runs for CMIP5 models taken from Geoffroy et al. (2013b). $\sigma_Q$ values are calculated from detrended $T(t)$ for either historical or piControl runs.

| Model | $C$ (W yr m$^{-2}$ K$^{-1}$) | $C_0$ (W yr m$^{-2}$ K$^{-1}$) | $\gamma$ (W m$^{-2}$ K$^{-1}$) | $\tau_f$ (yr) | $\tau_s$ (yr) | $a_f$ | $a_s$ | historical $\sigma_Q$ (W m$^{-2}$) | piControl $\sigma_Q$ (W m$^{-2}$) |
|---|---|---|---|---|---|---|---|---|---|
| ACCESS-CM2 | 7.9 | 88 | 0.59 | 6.1 | 286 | 0.51 | 0.49 | 0.65 | 0.43 |
| ACCESS-ESM1-5 | 7.0 | 87 | 0.70 | 5.0 | 255 | 0.47 | 0.53 | 0.47 | 0.43 |
| AWI-CM-1-1-MR | 7.3 | 52 | 0.54 | 4.2 | 143 | 0.67 | 0.33 | 0.56 | 0.46 |
| BCC-CSM2-MR | 8.0 | 68 | 0.73 | 4.6 | 165 | 0.55 | 0.45 | 0.63 | 0.56 |
| BCC-ESM1 | 7.6 | 85 | 0.62 | 4.9 | 235 | 0.58 | 0.42 | 0.53 | 0.43 |
| CESM2-WACCM | 6.6 | 82 | 0.81 | 4.5 | 237 | 0.41 | 0.59 | 0.59 | 0.47 |
| CIESM | 7.7 | 70 | 0.76 | 5.2 | 203 | 0.44 | 0.56 | 0.45 | 0.41 |
| CMCC-CM2-SR5 | 7.6 | 61 | 0.57 | 4.7 | 169 | 0.62 | 0.38 | 0.68 | 0.77 |
| CanESM5 | 7.7 | 74 | 0.54 | 6.3 | 259 | 0.52 | 0.48 | 0.55 | 0.42 |
| E3SM-1-0 | 7.8 | 40 | 0.41 | 7.4 | 169 | 0.55 | 0.45 | 0.52 | 0.39 |
| EC-Earth3 | 7.5 | 40 | 0.49 | 5.8 | 137 | 0.58 | 0.42 | 0.67 | 0.43 |
| EC-Earth3-Veg | 7.0 | 39 | 0.51 | 5.4 | 130 | 0.57 | 0.43 | 0.58 | 0.43 |
| FIO-ESM-2-0 | 7.2 | 86 | 0.79 | 4.4 | 217 | 0.49 | 0.51 | 0.54 | 0.41 |
| GFDL-CM4 | 5.3 | 84 | 0.69 | 3.7 | 245 | 0.49 | 0.51 | 0.48 | 0.35 |
| GFDL-ESM4 | 8.3 | 127 | 0.61 | 4.3 | 306 | 0.67 | 0.33 | 0.85 | 0.60 |
| GISS-E2-1-G | 6.4 | 143 | 0.88 | 2.7 | 264 | 0.61 | 0.39 | 0.92 | 0.73 |
| GISS-E2-1-H | 8.4 | 83 | 0.65 | 4.6 | 203 | 0.62 | 0.38 | 0.64 | 0.55 |
| HadGEM3-GC31-LL | 7.8 | 72 | 0.55 | 6.6 | 259 | 0.50 | 0.50 | 0.53 | 0.43 |
| HadGEM3-GC31-MM | 8.2 | 71 | 0.64 | 6.4 | 234 | 0.46 | 0.54 | 0.52 | 0.45 |
| IITM-ESM | 9.3 | 157 | 0.74 | 3.6 | 299 | 0.71 | 0.29 | 0.89 | 0.62 |
| INM-CM4-8 | 5.1 | 28 | 0.94 | 2.1 | 51 | 0.57 | 0.43 | 0.42 | 0.32 |
| INM-CM5-0 | 7.9 | 46 | 0.55 | 3.8 | 115 | 0.71 | 0.29 | 0.49 | 0.41 |
| IPSL-CM6A-LR | 7.2 | 61 | 0.46 | 6.0 | 222 | 0.59 | 0.41 | 0.47 | 0.45 |
| MIROC-ES2L | 11.4 | 232 | 0.63 | 5.2 | 521 | 0.71 | 0.29 | 1.12 | 0.89 |
| MIROC6 | 8.8 | 171 | 0.66 | 4.1 | 378 | 0.68 | 0.32 | 0.86 | 0.74 |
| MPI-ESM-1-2-HAM | 8.7 | 103 | 0.69 | 4.3 | 230 | 0.64 | 0.36 | 0.68 | 0.68 |
| MPI-ESM1-2-HR | 7.4 | 83 | 0.71 | 3.8 | 188 | 0.61 | 0.39 | 0.60 | 0.52 |
| MPI-ESM1-2-LR | 8.4 | 99 | 0.67 | 4.1 | 224 | 0.66 | 0.34 | 0.82 | 0.63 |
| MRI-ESM2-0 | 6.8 | 90 | 1.08 | 3.1 | 171 | 0.48 | 0.52 | 0.65 | 0.51 |
| NESM3 | 5.7 | 100 | 0.47 | 4.5 | 343 | 0.61 | 0.39 | 0.54 | 0.38 |
| SAM0-UNICON | 6.3 | 99 | 0.84 | 3.4 | 219 | 0.53 | 0.47 | 0.59 | 0.46 |
| TaiESM1 | 7.9 | 91 | 0.68 | 5.1 | 245 | 0.53 | 0.47 | 0.60 | 0.50 |
| UKESM1-0-LL | 7.4 | 75 | 0.54 | 6.0 | 259 | 0.53 | 0.47 | 0.63 | 0.44 |
| Multimodel mean | 7.6 | 87 | 0.66 | 4.7 | 230 | 0.57 | 0.43 | 0.63 | 0.51 |
| Standard deviation | 1.2 | 41 | 0.15 | 1.2 | 84 | 0.08 | 0.08 | 0.16 | 0.13 |

**Table B6.** Two-box parameters fitted from abrupt4xCO2 runs for CMIP6 models. $\sigma_Q$ values are calculated from detrended $T(t)$ for either historical or piControl runs.