# Peer review of "Testing the assumptions in emergent constraints: Why does the 'Emergent constraint on equilibrium climate sensitivity from global temperature variability' work for CMIP5 and not CMIP6?"

_EGUsphere, 2023_

## Author Response (AR1)

**Author response to referee comments and changes to 'Testing the assumptions...'**

February 2024

Dear Editor,

We would again like to thank the two reviewers for giving up their time to read our manuscript and for the positive evaluation.

We would also like to thank them for their suggestions and comments which we have addressed point-by-point below in bold face print. We have also revised the manuscript based on those suggestions.

During revisions we also found a small inconsistency in the way $\sigma_Q$ was calculated for the piControl runs compared to the historical runs. We have now corrected this. Some of the correlations have changed by a very small amount as a result but this has not changed any of our conclusions.

On behalf of all coauthors,

Mark Williamson

**1 RC1: BB Cael**

This manuscript addresses an important issue with a discrepancy between an emergent constraint that held well for CMIP5 and less so for CMIP6. The authors identify the reason, relating to an assumption in the emergent constraint's derivation.

I found this to be a very useful, interesting, and convincing paper; I cannot remember the last time, if ever, that I had so few comments on a manuscript. Nice work.

**Thank you for the comment, this is much appreciated.**

Minor Comments:

1.1 It would have been nice to explain a bit more in the abstract what 'the internal variability parameter' is, if there is space.

**We have updated that part of the abstract to read 'We show one assumption, that of low correlation and variation between ECS and the internal variability parameter, a parameter that captures chaotic internal variability as well as sub-annual (fast) feedbacks, while true for CMIP5 is not true for CMIP6. When accounted for, an emergent relationship appears once again in both CMIP ensembles implying the theoretical basis is still applicable although the original assumption in CHW18 is not.'**

1.2 It would have been nice to see Spearman or Kendall correlations & also p-values on these correlations, though I understand why the authors chose to stick with the Pearson correlation that's in the literature being compared to.

**We have calculated Spearman and Kendall correlations for all the results in the manuscript and they give essentially the same information as the Pearson correlation. This is because all relationships in the scatter plots are linearly and monotonically related both in theory and in the data. We have also calculated $p$ values for the Pearson correlations (given in the plot title) and adopted Schlund et al. (2020)'s definition of significance of a result given the $p$ value. These arguments have strengthened our results and we thank the referee for suggesting this.**

Even More Minor Comments:

1.3 Line 2 - suggest deleting '(then latest) state-of-the-art' - sounds impartial

**Removed.**

1.4 Line 7 - would say 'is weaker' not 'got weaker'

**Good suggestion, changed.**

1.5 Line 64 - incomplete sentence. In general lines 64-69 could use some rephrasing.

**We have rephrased this paragraph for clarity. It now reads 'The central interest of this manuscript is to test the assumptions that go into the derivation of the emergent relationship in CHW18. These assumptions are outlined in section 3 and then tested in the CMIP5 and CMIP6 model ensembles with the aim of understanding why the**

emergent relationship in CHW18 is weaker for the CMIP6 model ensemble. Of course all assumptions will be ultimately wrong if perfect agreement is expected (the often used quote 'all models are wrong' applies). However, 'some models are useful' and we look for agreement 'for all practical purposes (FAPP)', a term coined by John Bell (Bell, 1990). We will largely not be interested in the final step of obtaining the emergent constraint that results from combining the emergent relationship with observations for reasons we will outline later in the manuscript.'

1.6 Line 161 - uncorrelated with what?

**We have edited the sentence to read 'For both ensembles $Q_{2 \times CO2}$ is uncorrelated to ECS.'**

1.7 Line 268 - would replace 'good' with 'sound'

**Replaced.**

1.8 Line 272 - would replace 'good' with 'strong'

**Replaced.**

Thanks for the enjoyable read.

B. B. Cael

**2 RC2: Anonymous**

2.1 Overall, I acknowledge that the work has a robust, logical structure: the methodology is well established, some important additions, such as testing the 2-boxes conceptual model provide better context for the results obtained with the simpler Hasselmann's model. I do think, as well, that the manuscript falls a bit short in providing an explanation for the reasons why CMIP6 models do not exhibit such a strong emergent constraint as the one found in CMIP5 ensembles. In doing so, I struggle to see how the work could contribute to broader discussions on what use one can make of emergent constraints in the context of detection of the forced signal and attribution.

**We would argue that the central point of the manuscript is that we *do* provide a reason for why CMIP6 models do not exhibit as strong correlations as CMIP5 models, at least in the case for the emergent constraint presented in CHW18 i.e. a central assumption that held in the theory for the emergent relationship for the original CHW18 theory that held for CMIP5 no longer holds for CMIP6, namely the internal variability parameter being uncorrelated to ECS.**

Therefore, I think that the manuscript could substantially improve if at least the following two aspects would be taken into account.

2.2 Why is the internal variability parameter so crucial in defining the ECS emergent constraint in CMIP6. Why is it not the case for CMIP5?

**Again, we would argue the answer to this question is one of the main points, if not the main point of the manuscript. To repeat the argument again here: The equation used to derive the emergent relationship that the emergent constraint is founded on in CHW18 is a proportionality between ECS and the statistic of variability $\Psi$. However, there are some extra parameters, the internal variability parameter being one of them, that were treated as constants in the original study of CHW18 when only the CMIP5 ensemble was available. In that study and here in this manuscript, we show that this is a good assumption for CMIP5. However, we also show this is not a good assumption for CMIP6 - the internal variability parameter *is* correlated to ECS in this ensemble and *cannot* be treated as a constant. We demonstrate that when the internal variability parameter is incorporated into the predictor for the emergent constraint, good emergent relationships emerge in both CMIP5 and CMIP6 ensembles. We therefore identify why the emergent constraint in CHW18 in its original form failed in CMIP6 - it was not due to the theory being incorrect, it was due to the assumptions in applying the theory to the full complexity CMIP6 ensemble.**

2.3 Is the $ECS - \Psi$ constraint peculiar? What happens with other notable emergent constraints? Do they also differ from CMIP5? If so, do they differ because of similar reasons?

**Schlund et al. (2020) tested 11 emergent constraints found in CMIP5 and nearly all of these got weaker in CMIP6. We do not know whether they failed for similar reasons. Indeed, many of them do not have a simple theoretical model as a basis for their emergent relationship so assumption testing, the approach we follow in this manuscript, would be difficult to do. This is why we argue that emergent constraints should be based on a testable, falsifiable theoretical model. This aids understanding and lifts emergent constraint research from looking for strong correlations between variables to a more scientific approach of testing hypotheses of how the Earth system works. We have added some discussion along the lines of the referee's question in the discussion.**

2.4 I believe that the first point (2.3) is of particular relevance, given its implications for the development of synthetic model diagnostics and the usage of historical/paleoclimatic evidences to better understand and predict future climate scenarios.

**We agree. Looking at all the other emergent constraints and identifying why they got weaker in CMIP6 would be very beneficial to understanding and useful to the community. This however, would be a very large task and it is not clear (at least to us) how that could be done. The approach followed in this manuscript (testing the theoretical basis) would not be easy to apply to many other emergent constraints.**

SPECIFIC COMMENTS

2.5 56-57: some reference on the usage of simple models for the reproduction of forced global mean temperature response could be useful here and elsewhere in the manuscript;

**Good point. We have added Caldeira & Myhrvold (2013), Geoffroy et al. (2013a), Geoffroy et al. (2013b), Gregory (2000), Held et al. (2010) and MacMynowski et al. (2011).**

2.6 101-104: I do not have clear if the choice of 15 CMIP5 models is just guided by the need for consistency with Geoffrey et al. 2013 work or if there are other practical/theoretical reasons for that. Given that to the best of my knowledge more CMIP5 models should be available, I wonder if it would be possible to have a similar amount of models in both CMIP5 and CMIP6 ensembles. If that is not possible, I wonder if some arguments could be provided on the implications of the size of the ensemble for the robustness of the discussed relation.

We agree this could have been made clearer and we have added additional text in the revised manuscript. The reason is given in lines 101-104 (first submission) but more needed adding here. We analyze the same CMIP5 models as chosen in Geoffroy et al. (2013b) as we use their published parameter values in section 6 (new manuscript) to run simulations of the simple box models to compare the pure theory with the full complexity CMIP5 models. To fairly compare the full complexity model results with the box model results limits us to the same set of models - the set of 15 models Geoffroy et al. (2013b) chose to analyze.

For both CMIP5 and CMIP6, larger numbers of models (provided they are truly independent of each other) would give more robust emergent relationships. As outlined in lines 104-107 (first submission), for CMIP6 we chose the largest number of models that perform historical, piControl, abrupt-4xCO2 experiments for maximal robustness. Ideally of course, we would have many more (independent) models. We have now included $p$ values along with Pearson correlation $r$ in the titles of the plots. We have adopted Schlund et al. (2020)'s definition of significance based on $p$ value. $p$ values depend on the size of the sample (number of models in this case). Ensemble size is therefore implicitly included in the significance of the results. This has strengthened our original conclusions.

2.7 105-106: a bit in line with my previous comment, I see that it is a common procedure limiting to r1i1p1 or r1i1p1f1 runs, but it might be worthwhile, given that the interpretation strongly relies on the retrieval of the internal variability parameter, to discuss a bit if such parameter holds across different ensemble runs. It is not entirely straightforward to me, whether the choice of the parameter in the chosen runs would be representative of the other runs as well;

This is a good point and one we have thoroughly investigated. For models with multiple runs, we have drawn at random one run (r*i*p* or r*i*p*f* for CMIP5 and CMIP6 respectively) for each model and repeated the analysis in the manuscript multiple times. For the piControl runs, there are 15 and 24 unique permutations for the same set of CMIP5 and CMIP6 models respectively. The relatively low number of unique permutations is due to the low number of repeated runs for the piControl experiment. We have run the same analysis in this manuscript i.e. calculated the Pearson correlation, $r$, for every one of these permutations. For the historical runs, many more models do repeated runs multiple times. There are $1.6 \times 10^{12}$ and $5.4 \times 10^{20}$ unique permutations for the same set of CMIP5 and CMIP6 models for this experiment respectively. These numbers are clearly too large to search exhaustively. We have therefore drawn 1000 unique permutations for the historical experiment and repeated the analysis in this manuscript. The results are shown in two tables in the

revised manuscript under section 5 'Robustness to choice of model run'.

**The short answer is that the results reported for r1i1p1 or r1i1p1f1 are largely representative of a random run choice and the results do not change. A more detailed answer is given in the new section in the revised manuscript.**

2.8 155-156: provided the discussion above, I do not have clear why the authors opt for linearly detrending the temperature with the 55 years moving window, especially given that in a previous paper (Cox et al. 2018b) they noticed that retaining the external forcing would possibly improve the emergent relationship;

**We thank the referee for pointing this out. This was the procedure introduced in CHW18 and we continue with the same procedure here for consistency and comparison. The reasons for using a 55 year window have been discussed in the original paper (Cox et al. 2018a) as well as subsequent publications (Cox et al. 2018b and Williamson et al. 2018). The reason for linear detrending is to remove the response due to the slow timescale in the climate. It turns out that when fitting two-box models to the CMIP models, a fast timescale ($\sim 4$ years) and a slow timescale ($\sim 200$ years) response result, see Geoffroy et al. (2013b) for example or the tables in the appendix of the present manuscript. Linear detrending with a 55 year timescale fits nicely between the short and fast timescale and removes the slow response component. It also minimizes the uncertainty in the resulting emergent constraint (Cox et al 2018a). Removing the slow timescale response leaves a signal that is more Hasselmann (one-box) model like and therefore more like the underlying simple theory of the emergent relationship. We have expanded the paragraph starting at line 145 (first submission) to include this discussion which we neglected in the original manuscript.**

2.9 160: if the correlation value is meant to be the one in the title of the panels of in Figure 2, it would be useful to explicitly mention it here;

**Good suggestion. We have now changed the sentence at line 160 to read '... we plot $Q_{2\times CO2}$ against ECS and compute their correlation in both CMIP5 and CMIP6 ensembles. For both ensembles $Q_{2\times CO2}$ is uncorrelated to ECS ($r = -0.17$ for CMIP5 and $r = -0.07$ for CMIP6).'**

2.10 Figures 4 and 6: these figures, showing the relations between the two assumed parameters in CMIP5 and CMIP6, do not seem to add more arguments to the discussion than what already mentioned in the text. Consider whether is possible to remove them;

**We have removed them in the revised manuscript.**

2.11 Figures 9 and 10: when comparing the PDFs for piControl and historical, the authors evidence their similarities. It is a bit overlooked, though, that at first glance CMIP5 and CMIP6 in the historical runs exhibit substantially different medians and variance. Furthermore, the median for piControl in CMIP6 is possibly negative, whereas it is positive in CMIP5. Can the authors provide an explanation for that?

**Figs 9 and 10 (first submission, figs 7 and 8 new manuscript): Median and variance in CMIP5 and CMIP6 historical runs - Remember the histograms represent repeated simulations of the CMIP H76 emulators whereas the vertical dotted lines show the actual CMIP full complexity models. The idea of these figures is to show that the full complexity models can be reasonably well simulated by the pure theory (coloured dotted lines can be compared to the medians of the same coloured histogram). Agreement is reasonable and plausible (although much better for the two-box emulators in figs A1 and A2). The larger variance for the CMIP5 H76 emulator compared to the CMIP H76 emulator could be due to (i) the smaller number of models in the CMIP5 ensemble and/or (ii) a larger spread of $\sigma_Q$ values in CMIP5. Looking at tables B4 and B5 in the appendix, CMIP5 historical $\sigma_Q$ values have a larger standard deviation (0.18 versus 0.16 in CMIP6) confirming (ii) as a possible reason.**

**On 'Fig 9 (first submission, fig 7 new submission) LH panel $r(\Psi, ECS)$ median for piControl in CMIP6 is possibly negative, whereas it is positive in CMIP5. Can the authors provide an explanation for that?': The median of the correlation coefficient is negative but not significantly different from zero correlation for CMIP6. This result essentially backs up the central finding of the paper - that $\sigma_Q$ needs to be included i.e. $\Psi \rightarrow \Psi/\sigma_Q$ (RH panel of figure 9) to recover a significant correlation with ECS. This is true of the pure theory (histograms) or the full complexity models (dotted vertical line). Why could the median be at slightly negative correlation values between $\Psi$ and ECS (although the figure suggests not significantly different from zero correlation)? I guess this could be due to the slightly negative correlation between the internal variability parameter and ECS.**

2.12 288: the authors acknowledge that this remains an unanswered question, but I do think it is crucial to try to provide even a speculative explanation for that, in order to improve the usability of the main result described in the manuscript, at least something that could serve as triggering hypothesis for future work;

**We agree that it would be good to provide a route to answering this question and this is what the lines from 295-303 (first submission) do without speculating on why there is correlation between $\sigma_Q$ and ECS in CMIP6 (or lack of in CMIP5). We prefer not**

to speculate at this stage as it would be just hand waving.

TECHNICAL CORRECTIONS

2.13 67: for 'all practical purposes' → 'for all practical purposes';

**Thank you, good catch. Changed.**

2.14 175: is it actually $-0.06$, rather than $-0.60$;

**No, this was originally correct i.e. $r(\sigma_Q, ECS) = -0.6$ for CMIP6 piControl experiment showing that $\sigma_Q$ is still correlated to ECS in the piControl experiments as well as the historical. This is the opposite to CMIP5 that shows little correlation. However, as mentioned at the top of this reply we found a small inconsistency in the calculation of $\sigma_Q$ for the piControl run. Correcting this has meant $r(\sigma_Q, ECS) = -0.6 \rightarrow r(\sigma_Q, ECS) = -0.58$.**

2.15 226: 'results' → 'result';

**Changed.**

2.16 238: 'RH' → 'rhs';

**RH is the abbreviation of 'right hand'. We prefer to keep this as the sentence reads better that way.**

2.17 Figures 9 and 10: the legends do not seem to agree with the caption and the text;

**Thank you for pointing this out. We have corrected this.**

---

## Author Response (AR2)

**Author response to second round of referee comments and changes to 'Testing the assumptions...'**

April 2024

Dear Editor,

We would again like to thank all three reviewers for giving up their time to read our manuscript and for the positive evaluation. All three now recommend the manuscript for publication.

We would also like to thank them for their suggestions and comments. We have addressed the remaining referee comments (RC3) point-by-point below in bold face print. We have also revised the manuscript based on those suggestions.

Regarding RC2's question on the different role of the internal variability parameter between CMIP5 and CMIP6 we have added to the discussion in the revised manuscript.

On behalf of all coauthors,

Mark Williamson

**1 RC2: Anonymous**

I acknowledge that the manuscript

"Testing the assumptions in emergent constraints; Why does the 'Emergent constraint on equilibrium climate sensitivity from global temperature variability' work for CMIP5 and not CMIP6? by Mark S. Williamson, Peter M. Cox, Chris Huntingford and Femke J. M. M. Nijsse

has been substantially improved as compared to the previous version, and that most of the points raised in my comments have been addressed.

I do think, though, that the reason for such a different role of the internal variability parameter in CMIP5 and CMIP6 landscapes is somehow left untackled, and understand that the authors prefer not to speculate about that and commit to investigate into it in a future work.

I do think that this lack of interpretation hinders the relevance of the work, making it a bit dependent on the assumptions made and constraining the implications of the findings for future work, but I do acknowledge that this is not a justification for preventing the manuscript for being published.

Therefore, I leave this as a comment and urge the authors to provide an explanation for what has been found based on further evidences to be included in a future work, and recommend the manuscript to be accepted as is.

**We have added to the discussion the following paragraph:**

**'The question of why $\sigma_Q$ is correlated to ECS in CMIP6 and not CMIP5 is also left unanswered. However, one can speculate why this may be the case: As previously mentioned $\sigma_Q$ is a fitting parameter that is designed to capture the effect of chaotic internal variability as well as sub-annual (fast) feedbacks on global mean temperature variability. Zelinka et al. (2020) showed that the increased range of ECS in the CMIP6 models could be explained by the increased range in cloud feedbacks (see also Bock and Lauer (2024)). As $\sigma_Q$ is fitted to annual temperature timeseries, some of this fast (sub-annual) cloud feedback effect could be included in $\sigma_Q$ correlating it to ECS. We leave concrete answers to a future study.'**

**2 RC3: Anonymous**

This manuscript is well-motivated, well-written, and interesting. I appreciate the authors effort in investigating this curious issue and the thoroughness of the analysis. My comments are generally minor suggestions that are meant to clarify or improve the presentation to the reader. I recommend the manuscript for publication.

**Thank you for the positive review and the recommendation of publication.**

3.1 Lines 7 / 52: Schlund et al find an $r^2$ value of 0.01 in CMIP6. The language here suggests that there was still a (weaker) constraint, but it seems like the constraint (as used in Cox et al) had disappeared / was non-existent in CMIP6.

**To quote section 3.2 of Schlund et al (2020) 'a likely ECS range of 3.03 K $\pm$ 0.71 K for CMIP5 ($r^2 = 0.31$) and 3.44 K $\pm$ 1.15 K for CMIP6 ($r^2 = 0.08$)' for the CHW18 constraint. These correspond to $r = 0.56$ and $r = 0.28$ respectively comparable to the $r = 0.66$ and $r = 0.31$ found in this manuscript. Further on on section 3.2 they say the constraint is 'highly significant for the CMIP5 ensemble ($p = 0.0010$), but only almost significant for the CMIP6 ensemble ($p = 0.0545$)' also comparable to the $p = 0.007$ (highly significant) and $p = 0.079$ (almost significant) found in this manuscript. So I think the language is fine unless the referee is reporting later findings that we are unaware of?**

3.2 Line 17: I think the goal is not just to make climate models look increasingly realistic but to make climate models that are actually more realistic representations of Earths climate

**We have replaced 'look' with 'are'.**

3.3 Lines 24 - 25: Isnt this the likely range?

**Good spot. We have replaced 'high confidence' with 'likely'.**

3.4 Line 54: Note that you havent actually explained the physical underpinnings of the emergent constraint yet, so you are asserting that it has reasonable physical principles here (or relying on the reader to be familiar with Cox et al 2018).

**This is true. We do justify this assertion later on in the text and in previous manuscripts.**

3.5 Line 61: Is it true that temperature observations are relatively un-autocorrelated? I assume you mean in time? This strikes me as untrue. Could this be qualified to make it true? Or, alternatively, you could point to evidence that the observations are sufficiently reliable.

Yes, as the constraint is based on global mean temperature variability, we meant in time. Autocorrelation functions have $e$-folding times of a few years in the models and observations. While still not particularly precise we have changed to '...relatively un-autocorrelated in time (a few years) giving good estimators of the true values.' as the sentence is supposed to be introductory and adding too many details here makes it overly cumbersome.

3.6 Line 89: Should this be designed to model the *impact of* random, fast internally

Similar, but thank you. Good spot. We have changed this to 'fast (sub-annual), chaotic weather forcing on the slower Earth system components.'

3.7 Line 95: The CMIP5 historical experiment ended in 2005. Were these extended with another experiment?

Yes. This is explained in the following paragraph.

3.8 Lines 96 - 97: Im not sure where it would make sense to add this, but some studies indicated that the correlation coefficient was lower when more CMIP5 models were added to the original Cox et al. (2018) analysis (Po-Chedley et al. comment showed a correlation of r = 0.54 and Schlund et al. showed a correlation of r = 0.56). This slightly complicates the story a little bit (it isnt solely the CMIP6 ensemble that showed a weaker constraint). You might be able to allude to this in the paragraph at lines 113 to 116 (and you could note that $\sigma_Q$ improves the emergent relationship in both CMIP5 and CMIP6 later on in the manuscript).

We have added the following paragraph 'Although there were more CMIP5 models available than the $n = 16$ used in CHW18, the choice of one model per modelling centre was made to avoid biasing the emergent constraint towards similar models. Where multiple models were available from the same centre, the model with the lowest root mean square error to the observational temperature record was chosen. Po-Chedley et al. (2018) and Schlund et al. (2020) repeated the analysis of CHW18 including these additional models and thus had a larger CMIP5 ensemble (larger $n$). They found the emergent relationship got slightly weaker, although it was still 'highly significant' in the language of Schlund et al. (2020).'

3.9 Eq. 4: I had thought that you might get $\sigma_Q$ by analyzing variations in the TOA flux. I now realize that would be problematic because TOA balance includes the climate response ($\lambda T$) and isnt a pure estimate of weather forcing? Regardless, was this formula in Cox et al (2018)? Or a follow-up Williamson et al. paper? If so, could you cite them? I dont have great physical intuition for this equation, so providing more context (in text or via citations) would be helpful.

We have tried getting a proxy on $\sigma_Q$ along the lines you have suggested without luck so far. This is discussed in the conclusion. Regarding equation 4, a similar formula is given in Williamson et al (2018) (expressed with slightly different although equivalent parameters). We would imagine it appears in multiple contexts regarding guided random walks, Brownian motion, Ornstein-Uhlenbeck processes etc etc. It's a consequence of solving the one-box model with white noise forcing. We have added the citation to Williamson et al (2018) here.

3.10 Line 191: Is this expected because this was the case in Cox et al? If so, you could signal this by noting: Consistent with Cox et al. (2018),

**Changed to 'Consistent with CHW18...'.**

3.11 Lines 195 - 196: Im a little unclear about what exactly this sentence is saying. Could you just explicitly write out that $Q_{2\times CO2}/\sigma_Q$ is correlated with ECS in CMIP6 (I assume this is what you mean)?

**Corrected.**

3.12 Figures 2 - 4: Would it make sense to combine these figures into a single figure (with 6 subplots)? They seem sufficiently related that it would be helpful to put them together. This comment could also apply separately to Figures 5 and 6.

**This is a good idea and we've done something similar to your suggestion, combining figs 2 and 3 into one figure as well as figs 5 and 6 into one figure.**

3.13 Lines 201 - 203: It would be useful to plot the right hand side of Eq. 2 versus ECS and the 1-to-1 line so that the reader can see how accurately this equation predicts ECS in CMIP models (to be clear, I dont expect it to be perfect, but this would be a natural plot to include somewhere).

**A similar plot was in the original submission (fig 6) but was removed on the request of RC2 in the first revised submission. We have looked at these plots again, plotting ECS against $\frac{Q_{2\times CO2}}{\sigma_Q}\Psi$. As this relation should hold for all models running any experiment (provided you can remove the forced signal), we used an ensemble composed of all CMIP5 and CMIP6 models running both piControl and historical experiments (a total of 96 data points) and while the proportionality, $k$, in $ECS = k\frac{Q_{2\times CO2}}{\sigma_Q}\Psi$, holds with a high correlation value ($r = 0.74$, $p < 0.001$), the constant of proportionality is $k \sim 2\sqrt{2}$ rather than the $\sqrt{2}$ predicted by H76 i.e. the theoretical prediction of ECS from H76 (equation 2) is biased low. Using the two-box values and its theoretical prediction for ECS (equation (23) in Williamson et al. (2018)) brings $k$ closer to the**

predicted value $k \sim 1.3\sqrt{2}$ with similar high correlation ($r = 0.76$, $p < 0.001$). The two-box model adds a second, longer timescale, however the box model prediction of ECS is still slightly low. The low constant of proportionality could be due to these simple models not having other timescales that the complex models surely have. We have added this discussion to section 4 of the revised manuscript and in the discussion and conclusion as well as figures.

3.14 Line 353: I suggest being more explicit: namely $\sigma_Q$ is correlated with ECS.

**Good suggestion, done.**

**References**

L. Bock and A. Lauer. Cloud properties and their projected changes in cmip models with low to high climate sensitivity. *Atmos. Chem. Phys.*, 24(3):1587–1605, 2024. ISSN 1680-7324. doi: 10.5194/acp-24-1587-2024. URL https://acp.copernicus.org/articles/24/1587/2024/.

Stephen Po-Chedley, Cristian Proistosescu, Kyle C. Armour, and Benjamin D. Santer. Climate constraint reflects forced signal. *Nature*, 563(7729):E6–E9, 2018. ISSN 1476-4687. doi: 10.1038/s41586-018-0640-y. URL https://doi.org/10.1038/s41586-018-0640-y.

M Schlund, A Lauer, P Gentine, S C Sherwood, and V Eyring. Emergent constraints on Equilibrium Climate Sensitivity in CMIP5: do they hold for CMIP6? *Earth System Dynamics Discussions*, pages 1–40, 2020. doi: 10.5194/esd-2020-49. URL https://esd.copernicus.org/preprints/esd-2020-49/.

M. S. Williamson, P. M. Cox, and F. J. M. M. Nijsse. Theoretical foundations of emergent constraints: relationships between climate sensitivity and global temperature variability in conceptual models. *Dynamics and Statistics of the Climate System*, 3(1):dzy006, 2018. URL https://doi.org/10.1093/climsys/dzy006.

Mark D. Zelinka, Timothy A. Myers, Daniel T. McCoy, Stephen Po-Chedley, Peter M. Caldwell, Paulo Ceppi, Stephen A. Klein, and Karl E. Taylor. Causes of higher climate sensitivity in cmip6 models. *Geophysical Research Letters*, 47(1):e2019GL085782–e2019GL085782, 2020. URL https://agupubs.onlinelibrary.wiley.com/doi/abs/10.1029/2019GL085782.